# Occ3D: A Large-Scale 3D Occupancy Prediction Benchmark for Autonomous Driving

**Xiaoyu Tian**[1*]   **Tao Jiang**[1,3*]   **Longfei Yun**[1]   **Yucheng Mao**[1]   **Huitong Yang**[4]

**Yue Wang**[2]   **Yilun Wang**[1]   **Hang Zhao**[1,3,4†]

[1]IIIS, Tsinghua University
[2]University of Southern California   [3]Shanghai AI Lab   [4]Shanghai Qi Zhi Institute

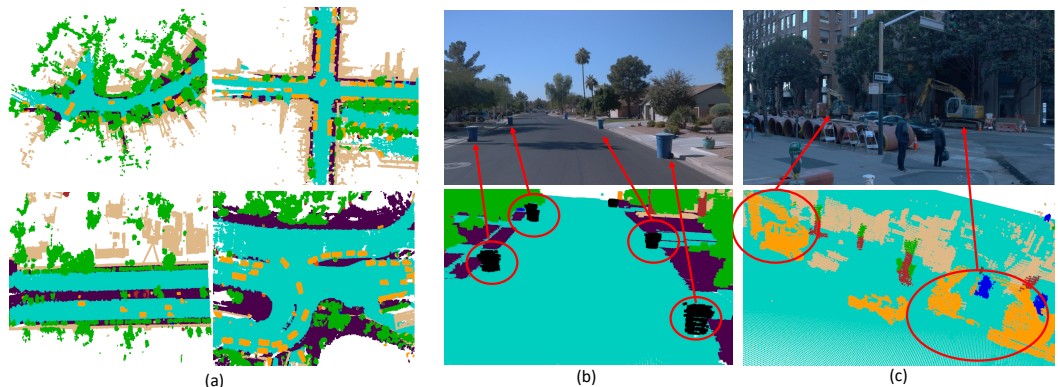

Figure 1: **Our Occ3D dataset demonstrates rich semantic and geometric expressiveness.** (a) Diversity of scenes in the Occ3D dataset; (b) Out-of-vocabulary objects, also known as General Objects (GOs), that cannot be extensively enumerated in the real world; (c) Irregularly-shaped objects that 3D bounding boxes fail to represent their accurate geometry.

## Abstract

Robotic perception requires the modeling of both 3D geometry and semantics. Existing methods typically focus on estimating 3D bounding boxes, neglecting finer geometric details and struggling to handle general, out-of-vocabulary objects. 3D occupancy prediction, which estimates the detailed occupancy states and semantics of a scene, is an emerging task to overcome these limitations. To support 3D occupancy prediction, we develop a label generation pipeline that produces dense, visibility-aware labels for any given scene. This pipeline comprises three stages: voxel densification, occlusion reasoning, and image-guided voxel refinement. We establish two benchmarks, derived from the Waymo Open Dataset and the nuScenes Dataset, namely Occ3D-Waymo and Occ3D-nuScenes benchmarks. Furthermore, we provide an extensive analysis of the proposed dataset with various baseline models. Lastly, we propose a new model, dubbed Coarse-to-Fine Occupancy (CTF-Occ) network, which demonstrates superior performance on the Occ3D benchmarks. The code, data, and benchmarks are released at https://tsinghua-mars-lab.github.io/Occ3D/.

---

*Authors contributed equally.

†Corresponding to: hangzhao@mail.tsinghua.edu.cn

37th Conference on Neural Information Processing Systems (NeurIPS 2023) Track on Datasets and Benchmarks.

# 1   Introduction

3D perception is a crucial component in vision-based robotic systems like autonomous driving. One of the most popular visual perception tasks is 3D object detection, which estimates the 3D locations and dimensions of objects defined in a pre-determined ontology tree [48, 22]. While the resulting 3D bounding boxes are compact, the level of expressiveness they provide is restricted, as illustrated in Figure 1: (1) 3D bounding box representation erases the geometric details of objects, a construction vehicle has a mechanical arm that protrudes from the main body; (2) uncommon categories, like trash cans on the streets, are often ignored and not labeled in the datasets [4, 43] since object categories in the open world cannot be extensively enumerated.

These limitations call for a general and coherent representation that can model the detailed geometry and semantics of objects both within and outside of the ontology tree. 3D Occupancy Prediction, *i.e.* understanding every voxel in the 3D space, is an important task to achieving this goal. We formalize the 3D occupancy prediction task as follows: a model needs to jointly estimate the *occupancy state* and *semantic label* of every voxel in the scene from images [2, 24, 5]. The occupancy state of each voxel can be categorized as *free*, *occupied*, or *unobserved*. For occupied voxels, semantic labels are assigned. For objects that are not in the predefined categories, they are labeled as *General Objects (GOs)*. Although GOs are rare, they are essential for perception tasks with safety considerations since they are typically undetected by 3D object detection with predefined categories.

Despite recent advancements in 3D occupancy prediction [5, 16, 53], there is a notable absence of high-quality datasets together with benchmarks. Constructing such a dataset is challenging due to three major issues: sparsity, occlusion and 3D-2D misalignment. To overcome these hurdles, we create a semi-automatic label generation pipeline that consists of three steps: voxel densification, occlusion reasoning, and image-guided voxel refinement. Each step within our pipeline is validated through a 3D-2D consistency metric, demonstrating that our proposed label generation pipeline effectively generates dense and visibility-aware annotations.

Building upon the public Waymo Open Dataset [43], nuScenes [4] and Panoptic nuScenes [11] Dataset, we produce two benchmarks for our task accordingly, Occ3D-Waymo and Occ3D-nuScenes. Compared to conventional datasets such as SemanticKITTI [2] and KITTI-360 [24], our Occ3D is the first dataset to offer the surround-view images and high-resolution 3D voxel occupancy representation with the most diverse scenarios.

A series of recent occupancy prediction models are reproduced and benchmarked on Occ3D. Additionally, we propose CTF-Occ, a transformer-based **C**oarse-**T**o-**F**ine 3D **Occ**upancy prediction network. CTF-Occ achieves superior performance by aggregating 2D image features into 3D space via cross-attention in an efficient coarse-to-fine fashion.

The contributions of this work are as follows: (1) We introduce Occ3D, a high-quality 3D occupancy prediction benchmark to facilitate research in this emerging area; (2) We put forward a rigorous automatic label generation pipeline for constructing the Occ3D benchmark, with comprehensive validation of the effectiveness of the pipeline; (3) We benchmark existing model and propose a new CTF-Occ network that achieves superior 3D occupancy prediction performance.

# 2   Related Work

**3D detection.** The goal of 3D object detection is to estimate the locations and dimensions of objects within a predefined ontology. 3D object detection is often performed in LiDAR point clouds [55, 18, 51, 52, 34, 36, 10, 39]. More recently, vision-based 3D object detection has gained more attention due to its low cost and rich semantic content [41, 46, 48, 22, 26, 33, 28, 13, 27, 15, 21, 31, 25]. Several LiDAR-camera fusion methods are also proposed [35, 8, 28].

**3D occupancy prediction.** A related task of 3D occupancy prediction is Occupancy Grid Mapping (OGM) [30, 44, 47], a classical task in mobile robots that aims to generate probabilistic maps from sequential noisy range measurements. OGM can be solved within a Bayesian framework, some recent works further combine semantic segmentation with OGM for downstream tasks [17, 42, 37]. Note that OGM requires range sensors, and also makes the assumption that the scene is static over time. The 3D occupancy prediction task does not have these constraints and can be applied in

Table 1: **Dataset comparison**. Comparing Occ3D Datasets with other occupancy prediction datasets. Surround = ✓ represents surround-view image inputs. C, D, L denote camera, depth and LiDAR.

| Dataset | Type | Surround | Modality | # Classes | # Sequences | # Frames | Volume Size | Resolution (m) |
|---|---|---|---|---|---|---|---|---|
| NYUv2 [40] | Indoor | ✗ | C & D | 11 | 464 | 1449 | [240, 240, 14] | - |
| ScanNet [9] | Indoor | ✗ | C & D | 11 | 1513 | 1513 | [62, 62, 31] | - |
| SemanticKITTI [2] | Outdoor | ✗ | C & L | 28 | 22 | 4,3000 | [256, 256, 32] | [0.2, 0.2, 0.2] |
| KITTI-360 [23] | Outdoor | Fisheye | C & L | 19 | 11 | 90,960 | [256, 256, 32] | [0.2, 0.2, 0.2] |
| **Occ3D-nuScenes** | Outdoor | ✓ | C & L | 16+$GO$ | 1000 | 40,000 | [200, 200, 16] | [0.4, 0.4, 0.4] |
| **Occ3D-Waymo** | Outdoor | ✓ | C & L | 14+$GO$ | 1000 | 200,000 | [3200, 3200, 128] | [0.05, 0.05, 0.05] |

vision-only robotic systems in dynamic scenes. Recently, TPVFormer [16] proposes a tri-perspective view method to predict 3D occupancy. However, its output is sparse due to LiDAR supervision.

**Semantic scene completion.** Another related task is Semantic Scene Completion (SSC) [1, 6, 9, 3, 24, 50, 38, 49, 7, 19, 50, 32, 20, 29, 54], whose goal is to estimate a dense semantic space from partial observations. SSC differs from 3D occupancy prediction in two ways: (1) SSC focuses on inferring occluded regions given visible parts, while occupancy prediction does not intend to estimate the invisible regions; (2) existing SSC task typically deals with static scenes, whereas occupancy prediction works with dynamic ones.

## 3 Occ3D Dataset

### 3.1 Task Definition

Given a sequence of sensor inputs, the goal of 3D occupancy prediction is to estimate the state of each voxel in the 3D scene. Specifically, the input of the task is a $T$-frame historical sequence of $N$ surround-view camera images $\{\mathcal{I}_{i,t} \in \mathbf{R}^{H_i \times W_i \times 3}\}$, where $i = 1, ..., N$ and $t = 1, ..., T$. We also assume known sensor intrinsic parameters $\{K_i\}$ and extrinsic parameters $\{[R_i|t_i]\}$ in each frame. The ground truth labels are the voxel states, including *occupancy state* ("occupied", "free", or "unobserved") and *semantic label* (category, or "unknow"). For example, a voxel on a vehicle is labeled as ("occupied", "vehicle"), and a voxel in the free space is labeled as ("free", None). Note that the 3D occupancy prediction framework also supports extra attributes as outputs, such as *instance IDs* and *motion vectors*; we leave them as future work.

### 3.2 Dataset Statistics

We generate two 3D occupancy prediction datasets, Occ3D-nuScenes and Occ3D-Waymo. Occ3D-nuScenes contains 600 scenes for training, 150 scenes for validation, and 150 for testing, totaling 40,000 frames. It has 16 common classes with an additional general object (GO) class. Each sample covers a range of [-40m, -40m, -1m, 40m, 40m, 5.4m] with a voxel size of [0.4m,0.4m,0.4m]. Occ3D-Waymo contains 798 sequences for training, 202 sequences for validation, accumulating 200,000 frames. It has 14 known object classes with an additional GO class. Each sample covers a range of [-80m, -80m, -1m, 80m, 80m, 5.4m], with an extremely fine voxel size of [0.05m, 0.05m, 0.05m].

Occ3D stands out when compared with other datasets, as shown in Table 1. The indoor datasets NYUv2 and ScanNet lack surround images and consist of fewer sequences and frames. SemanticKITTI and KITTI-360, the two other outdoor datasets, also lack surround images, with the exception of KITTI-360's fisheye images. For the safety of autonomous driving, the general object class is particularly important, a feature that is not available in the SemanticKITTI and KITTI-360 datasets. Furthermore, Occ3D-Waymo is currently the 3D occupancy dataset with the most diverse scenarios, comprehensive labels, and the highest resolution among all open-source datasets.

### 3.3 Dataset Construction Pipeline

Annotating 3D occupancy from images is impossible due to the lack of accurate depth and geometry. Therefore, we take advantage of LiDAR scans and their annotations to construct high-quality occupancy labels. However, there are three primary hurdles: **sparsity**, **occlusion**, and **3D-2D misalignment**. Sparsity refers to the fact that LiDAR scans are sparse, thereby hindering the acquisition

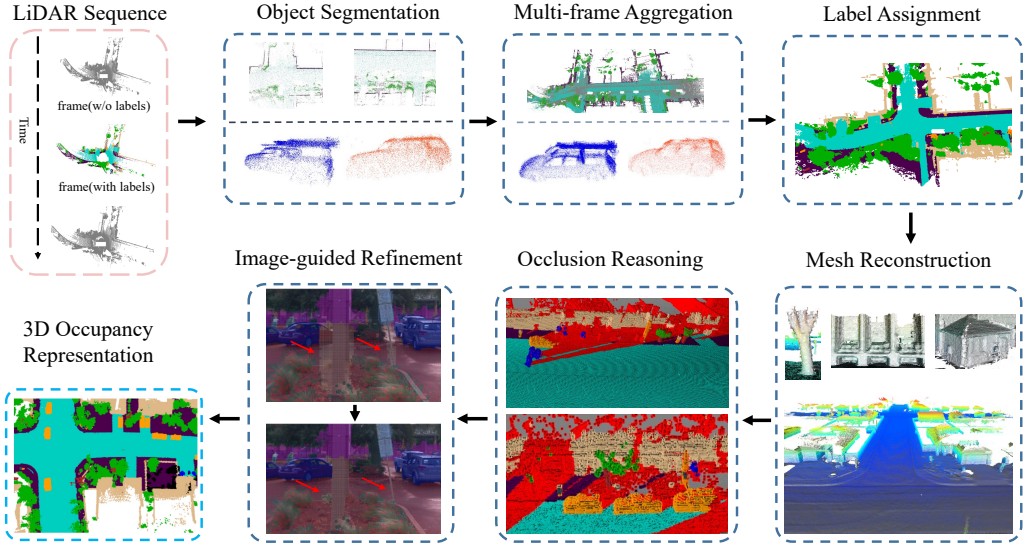

Figure 2: **Overview of the label generation pipeline.** The pipeline consists of three main steps: voxel densification, occlusion reasoning, and image-guided voxel refinement. Voxel densification consists of object segmentation, multi-frame aggregation, and label assignment.

of dense voxels. Occlusion, on the other hand, is concerned with the identification of voxels that, once densified, become invisible in the current image view due to occlusion. 3D-2D misalignment pertains to the disparities when projecting the 3D voxels onto 2D images, often induced by sensor noises or pose errors.

Our proposed label generation pipeline addresses the above challenges, an overview is shown in Figure 2. Initially, in **voxel densification**, we increase the density of the point clouds by performing multi-frame aggregation for both static and dynamic objects separately. Then we employ a K-nearest neighbor algorithm to assign labels to unlabeled points and utilize mesh reconstruction to perform hole-filling. Subsequently, we carry out **occlusion reasoning** from both LiDAR and camera perspectives, utilizing a ray-casting operation to label the occupancy state of each voxel. Finally, misaligned voxels are eliminated through an **image-guided voxel refinement** process. We provide pseudo-code and the hyper-parameters of each step in the Appendix.

### 3.3.1 Voxel Densification

LiDAR data is inherently sparse, to acquire dense point clouds: 1) We aggregate all points throughout the frames, treating dynamic objects and static background points separately; 2) We take advantage of unlabeled frames (which we'll refer to as non-keyframes) and use a K-Nearest Neighbors (KNN) algorithm to assign semantic labels; 3) In spite of frame aggregation, there persist holes on the object surfaces, we fill these holes with mesh reconstruction.

**Dynamic and static objects segmentation.** Point clouds derived from individual frames are categorized into "dynamic objects" and "static scenes". The static scenes contain entities such as ground, buildings, and road signs that do not exhibit positional change over time. Dynamic objects, such as cars and pedestrians need to be segregated since naive temporal aggregation results in motion blur.

**Multi-frame aggregation.** After segregating dynamic objects from static scenes, multi-frame aggregation is conducted separately on them. For dynamic objects, we extract the points located within the annotated or tracked box and subsequently transform them from sensor coordinates to box coordinates. By concatenating these transformed points, we densify the point cloud of dynamic objects. For the static scene, we simply aggregate its points across time in the global coordinate system. The static scene is then fused with the aggregated dynamic objects in the current frame, thereby generating a single-frame dense point cloud.

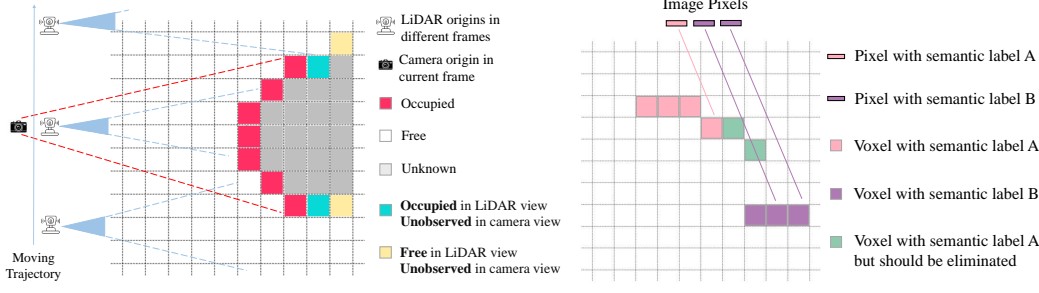

(a) Occlusion Reasoning for Visibility Mask        (b) Image-guided Voxel Refinement

Figure 3: **Visibility and refinement.** (a) LiDAR visibility: a voxel is "occupied" if it reflects LiDAR (red voxels), or "free" if it is traversed through by a ray (white voxels); Camera visibility: Any voxel not scanned by camera rays is set to "unobserved" (blue and yellow voxels). (b) Image-guided voxel refinement: during ray casting, when the first voxel with the same semantic label as the pixel label is encountered, we set the previously traversed voxel states to "free" (green voxels).

**KNN for label assignment.** The task of directly annotating each point in every frame is labor-intensive. Current datasets only annotate a selected portion of the frames - for instance, the Waymo dataset proceeds at a rate of 2Hz, whereas Lidar scans operate at a 10Hz frequency. To utilize the unlabeled frames, we employ the K-nearest neighbors (KNN) algorithm to assign semantic labels to each unlabeled point. Specifically, for each point in the unlabeled frame, we find the K nearest keyframe points and assign the majority semantic label.

**Mesh reconstruction.** After multi-frame aggregation, the density of point clouds is still not enough to produce high-quality dense voxels: a smaller voxel size may lead to objects with many holes, while a larger voxel size could induce excessive smoothness. To mitigate these issues, we perform mesh reconstruction. For non-ground categories, we optimize surfaces through VDBFusion [45], an approach for volumetric surface reconstruction based on truncated signed distance functions (TSDF). The flexibility and efficacy of VDBFusion surpass traditional methods such as Poisson surface reconstruction. For the ground, VDBFusion fails as small ray angles result in incorrect TSDF values. We instead establish uniform virtual grid points and fit each local surface mesh using points within a small region. After reconstructing the meshes, dense point sampling is performed, and KNN is further adopted to assign semantic labels to the sampling points.

### 3.3.2 Occlusion Reasoning for Visibility Mask

We perform occlusion reasoning and introduce LiDAR visibility mask and camera visibility mask to further enhance our 3D occupancy prediction benchmark.

**Aggregated LiDAR visibility mask.** To obtain a 3D occupancy grid from aggregated LiDAR point clouds, a straightforward way is to set the voxels containing points to be "occupied" and the rest to "free". However, since LiDAR points are sparse, some occupied voxels are not scanned by LiDAR beams, and can be mislabeled as "free". To avoid this issue, we perform a ray casting operation to determine the visibility of each voxel, as shown in Figure 3a. Concretely, we cast a ray from the sensor origins to each LiDAR point. A voxel is considered visible if it either reflects LiDAR points, or if it is traversed through by a ray. If neither condition is met, the voxel is classified as "unobserved".

**Camera visibility mask.** We connect each occupied voxel center with the camera origin, thereby forming a ray. Along each ray, we set the first occupied voxel as "observed", and the remaining as "unobserved". Any voxel not scanned by camera rays is set to "unobserved" as well. Determining the visibility of a voxel is crucial for the evaluation of the 3D occupancy prediction task: evaluation is only performed on the "observed" voxels in both the LiDAR and camera views.

### 3.3.3 Image-guided Voxel Refinement

Influences such as LiDAR noise and pose drifts can cause the 3D shape of objects to appear larger than their actual physical dimensions. To rectify this, we further refine the dataset by eliminating

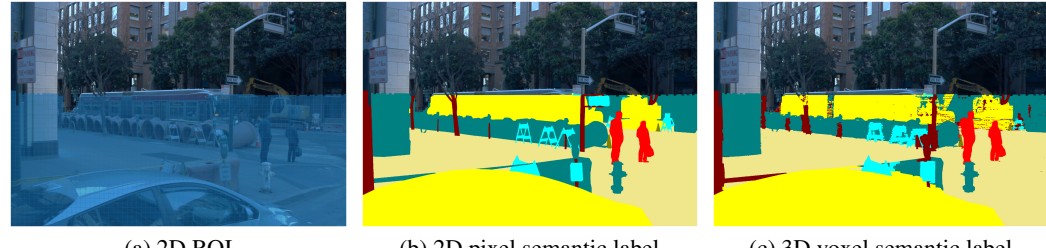

| (a) 2D ROI | (b) 2D pixel semantic label | (c) 3D voxel semantic label |

Figure 4: **3D-2D consistency** (a) 2D ROI within single-frame LiDAR scan range. (b) Semantic labels of a single image within the 2D ROI. (c) The reprojection of 3D voxel semantic labels onto the image within the 2D ROI.

incorrectly occupied voxels, guided by semantic segmentation masks of images. As shown in Figure 3b, to obtain the correspondence between 3D voxels and 2D pixels, we adopt a ray casting operation similar to the one in the previous section: connecting each occupied voxel center with the camera center to form a ray, and traverse the voxel that this ray passes through from near to far from the pixel origin. When the first voxel with the same semantic label as the pixel label is encountered, we set the previously traversed voxel states to "free". This step greatly improves the shape at object boundaries.

## 4   Quality Check

Acquiring an occupancy representation that adheres to the complete shape of all objects is challenging. Therefore, evaluating the quality of the dataset and ensuring the effectiveness of each step in our pipeline is critical. To this end, we propose a method that evaluates the quality of occupancy by checking semantic consistency between 2D pixels and their corresponding voxel.

### 4.1   3D-2D consistency

Compared to 3D occupancy semantic labels obtained through aggregation and reconstruction, 2D semantic masks manually annotated by humans are highly accurate. Thus, we assess the quality of the dataset by verifying the 3D-2D consistency between semantic labels of 3D voxels and their corresponding 2D image pixels. We calculate 3D-2D consistently in three steps: filtering the 2D pixel region involved in consistency calculation for the current frame, identifying the corresponding 3D voxels of this pixel region, and finally, computing their 3D-2D semantic consistency.

**2D ROI.** 2D images contain objects that are beyond the scanning range of the LiDAR sensor. When calculating 3D-2D consistency, we use the maximum range covered by a single LiDAR frame as the 2D Region of interest (ROI). Specifically, we project single-frame LiDAR points onto the 2D image coordinate system using LiDAR-to-camera transformation. Then, our algorithm traverses in the horizontal coordinate direction and selects the highest vertical coordinate of the projected points in each vertical column as the height of that column. As shown in Figure 4a, all pixels below this height are treated as the 2D valid region involved in the consistency calculation.

**3D label query.** After determining the 2D ROI in each image, we identify its corresponding 3D voxels for these regions. Since each voxel has a certain volume, directly projecting them onto a 2D image poses a multi-pixel association issue. Moreover, when the projection overlap occurs, determining the corresponding occlusion relationship becomes complicated. We instead query corresponding 3D voxels for each 2D image pixel. Specifically, for each pixel in the selected region, we perform ray traversal and find the closest 3D voxel to the ray.

**Metrics.** To evaluate the dataset quality, for each pixel in an image, we compare its semantic label with the semantic prediction of its corresponding 3D voxel. We adopt the standard Precision, Recall, Intersection-over-Union(IoU), and mean Intersection-over-Union(mIoU) metric.

Table 2: **Quantitative results for design choices.** *SFP*, single frame points; *MFP*, aggregating points from unlabeled frames; *VS*, short for voxel size; *Mesh*, showcasing mesh reconstruction; and *IGR*, denoting image-guided voxel refinement. The three numbers from top to bottom in each choice are IoU, recall, and precision for the specific class.

| SFP | MFP | VS | Mesh | IGR | vehicle | bicyclist | ped | sign | road | pole | cone | bicycle | building | mIOU |
|---|---|---|---|---|---|---|---|---|---|---|---|---|---|---|
| ✓ | | - | | | 5.87 | 5.12 | 3.65 | 3.47 | 0.33 | 0.10 | 0.09 | 0.11 | 0.34 | |
| | | | | | 8.53 | 6.61 | 4.81 | 3.85 | 0.33 | 0.10 | 0.09 | 0.11 | 0.34 | 13.32 |
| | | | | | 95.38 | 58.66 | 60.13 | 61.44 | 92.78 | 34.90 | 25.35 | 60.12 | 66.93 | |
| ✓ | ✓ | - | | | 37.89 | 37.99 | 28.25 | 12.57 | 11.70 | 5.48 | 3.51 | 6.01 | 15.49 | |
| | | | | | 40.02 | 58.77 | 37.21 | 14.80 | 12.06 | 6.32 | 3.99 | 6.45 | 17.69 | 17.65 |
| | | | | | 87.48 | 51.79 | 53.98 | 45.45 | 79.72 | 29.25 | 22.37 | 46.76 | 55.49 | |
| ✓ | ✓ | 0.1 | | | 75.23 | 38.66 | 30.78 | 33.77 | 56.30 | 30.58 | 24.03 | 31.36 | 49.85 | |
| | | | | | 91.20 | 87.00 | 60.90 | 56.80 | 67.35 | 55.97 | 42.22 | 37.00 | 68.66 | 41.17 |
| | | | | | 81.12 | 41.03 | 38.37 | 45.45 | 95.53 | 40.27 | 35.81 | 67.28 | 64.53 | |
| ✓ | ✓ | 0.1 | ✓ | | 75.13 | 37.97 | 30.36 | 32.88 | 82.79 | 16.63 | 17.48 | 32.46 | 52.27 | |
| | | | | | 90.98 | 83.02 | 55.07 | 54.93 | 85.34 | 63.57 | 58.13 | 40.77 | 77.81 | 41.99 |
| | | | | | 81.17 | 41.17 | 40.36 | 45.02 | 96.52 | 18.38 | 19.99 | 61.43 | 61.43 | |
| ✓ | ✓ | 0.05 | ✓ | | 78.76 | 46.33 | 34.04 | 34.85 | 64.56 | 18.28 | 20.57 | 42.59 | 52.27 | |
| | | | | | 89.20 | 84.22 | 55.42 | 52.02 | 67.43 | 63.17 | 58.89 | 53.71 | 84.73 | 43.58 |
| | | | | | 87.06 | 50.74 | 46.87 | 51.36 | 93.80 | 20.46 | 24.02 | 67.28 | 57.71 | |
| ✓ | ✓ | 0.05 | ✓ | ✓ | 88.82 | 76.89 | 47.54 | 50.18 | 71.97 | 31.77 | 32.09 | 66.40 | 60.90 | |
| | | | | | 91.34 | 92.12 | 63.78 | 61.20 | 75.11 | 73.80 | 62.44 | 77.67 | 94.73 | 58.50 |
| | | | | | 96.98 | 82.31 | 65.11 | 73.59 | 94.51 | 35.80 | 39.76 | 82.07 | 63.04 | |

## 4.2 Quantitative Results

Table 2 shows the performance gain of each proposed step of our auto-labeling pipeline. The 3D-2D consistency is evaluated in a subset of Occ3D-Waymo. Single-frame points (SFP) means that we only use a single-frame point cloud to calculate its 3D-2D consistency using the previously proposed method. As shown in the table, our method achieves high SFP precision and low recall. In addition to SFP, we aggregate points from multiple frames (MFP). Compared to SFP, MFP sees a significant improvement in recall, but its precision decreases to a certain extent, which is caused by the LiDAR noise and/or pose errors. Based on MFP, we study the effect of voxelization, which leads to better precision and recall. This further validates the effect of correction on pose inaccuracies. As mentioned before, a small voxel size results in objects containing many holes, while a larger voxel size leads to over smoothness. The former results in low recall, while the latter results in low precision. We use mesh reconstruction to alleviate the hole issue in objects caused by a small voxel size, which is reflected by the comparison between third row and fifth row in the table. Finally, we demonstrate that our proposed image-guided refinement indeed promotes the 3D-2D semantic consistency, shown in the last row.

## 5 Coarse-to-Fine Occupancy Network

To deal with the challenging 3D occupancy prediction problem, we present a new transformer-based model named **C**oarse-**t**o-**F**ine **Occ**upancy (CTF-Occ) network. An overview of CTF-Occ network is shown in Figure 5. First, 2D image features are extracted from multi-view images with an image backbone. Then, 3D voxel queries aggregate 2D image features into 3D space via a cross-attention operation. Our approach involves using a pyramid voxel encoder that progressively improves voxel feature representation through incremental token selection and spatial cross-attention in a coarse-to-fine fashion. This approach enhances the spatial resolution and refines the detailed geometry of objects, ultimately leading to more accurate 3D occupancy predictions.

**Incremental token selection.** The task of predicting 3D occupancy requires a detailed representation of geometry, but this can result in significant computational and memory costs if all 3D voxel tokens are used to interact with regions of interest in the multi-view images. Given that most 3D voxel grids

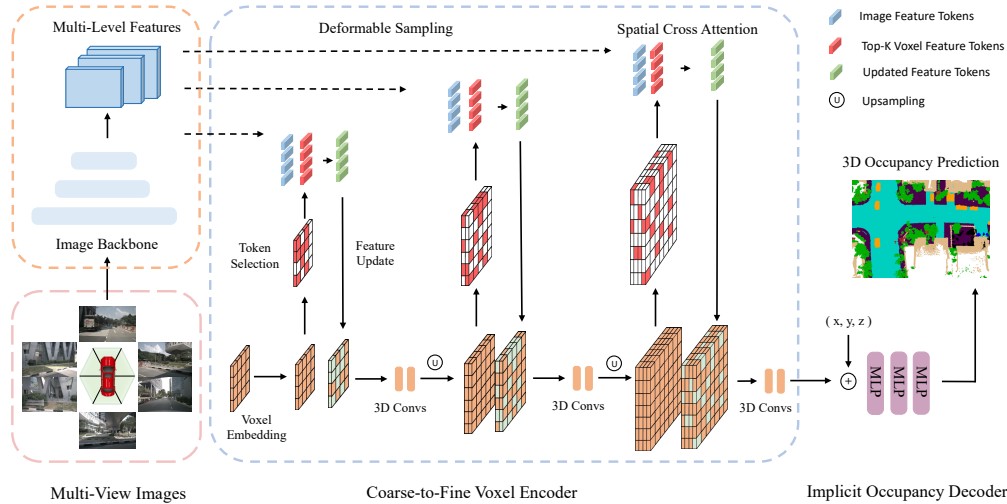

Figure 5: **The architecture of CTF-Occ network.** CTF-Occ consists of an image backbone, a coarse-to-fine voxel encoder, and an implicit occupancy decoder.

in a scene are empty, we propose an incremental token selection strategy that selectively chooses foreground and uncertain voxel tokens in cross-attention computation. This strategy enables adaptive and efficient computation without sacrificing accuracy. Specifically, at the beginning of each pyramid level, each voxel token is fed into a binary classifier to predict whether this voxel is empty or not. We use the binary ground-truth occupancy map as supervision to train the classifier. In our approach, we select the K-most uncertain voxel tokens for the subsequent feature refinement.

**Spatial cross attention.** At every level of the pyramid, we first select the top-K voxel tokens and then aggregate the corresponding image features. In particular, we apply 3D spatial cross-attention [22] to further refine the voxel features.

**Convolutional feature extractor.** Once we apply deformable cross-attention to the relevant image features, we proceed to update the features of the foreground voxel tokens. Then, we use a series of stacked convolutions to enhance feature interaction throughout the entire 3D voxel feature maps. At the end of the current level, we upsample the 3D voxel features using trilinear interpolation.

**Occupancy decoder.** The CTF voxel encoder generates voxelized feature output $V_{out} \in \mathbb{R}^{W \times H \times L \times C}$. Then the voxel features $V_{out}$ are fed into several MLPs to obtain the final occupancy prediction $O \in \mathbb{R}^{W \times H \times L \times C'}$, where $C'$ is the number of the semantic classes. Furthermore, we introduce an implicit occupancy decoder that can offer arbitrary resolution output by utilizing implicit neural representations. The implicit decoder is implemented as an MLP that outputs a semantic label by taking two inputs: a voxel feature vector extracted by the voxel encoder and a 3D coordinate inside the voxel.

# 6 Experiments

To benchmark our proposed Occ3D datasets and our CTF-Occ model, we evaluate existing 3D occupancy prediction methods on Occ3D-nuScenes and Occ3D-Waymo.

## 6.1 Experimental Setup

**Dataset and Metrics.** Occ3D-Waymo contains 1,000 publicly available sequences in total, where 798 scenes are for training and 202 scenes are for validation. The scene range is set from -40m to 40m along X and Y axis, and from -5m to 7.8m along Z axis. Occ3D-nuScenes contains 700 training scenes and 150 validation scenes. The occupancy scope is defined as -40m to 40m for X and Y axis, and -1m to 5.4m for the Z axis. We choose a voxel size of $0.4$m to conduct our experiments on both two datasets. We adopt the metrics of Intersection-over-Union (IoU) and mean Intersection-over-Union(mIoU) to evaluate performance.

Table 3: 3D occupancy prediction performance on the Occ3D-nuScenes dataset. Cons. Veh represents construction vehicle and Dri. Sur is for driveable surface.

| Method | others | barrier | bicycle | bus | car | Cons. Veh | motorcycle | pedestrian | traffic cone | trailer | truck | Dri. Sur | other flat | sidewalk | terrain | manmade | vegetation | mIoU |
|---|---|---|---|---|---|---|---|---|---|---|---|---|---|---|---|---|---|---|
| MonoScene [5] | 1.75 | 7.23 | 4.26 | 4.93 | 9.38 | 5.67 | 3.98 | 3.01 | 5.90 | 4.45 | 7.17 | 14.91 | 6.32 | 7.92 | 7.43 | 1.01 | 7.65 | 6.06 |
| TPVFormer [16] | 7.22 | 38.90 | 13.67 | 40.78 | 45.90 | 17.23 | 19.99 | 18.85 | 14.30 | 26.69 | 34.17 | 55.65 | 35.47 | 37.55 | 30.70 | 19.40 | 16.78 | 27.83 |
| BEVDet [14] | 4.39 | 30.31 | 0.23 | 32.26 | 34.47 | 12.97 | 10.34 | 10.36 | 6.26 | 8.93 | 23.65 | 52.27 | 24.61 | 26.06 | 22.31 | 15.04 | 15.10 | 19.38 |
| OccFormer [53] | 5.94 | 30.29 | 12.32 | 34.40 | 39.17 | 14.44 | 16.45 | 17.22 | 9.27 | 13.90 | 26.36 | 50.99 | 30.96 | 34.66 | 22.73 | 6.76 | 6.97 | 21.93 |
| BEVFormer [22] | 5.85 | 37.83 | 17.87 | 40.44 | 42.43 | 7.36 | 23.88 | 21.81 | 20.98 | 22.38 | 30.70 | 55.35 | 28.36 | 36.0 | 28.06 | 20.04 | 17.69 | 26.88 |
| **CTF-Occ (Ours)** | 8.09 | 39.33 | 20.56 | 38.29 | 42.24 | 16.93 | 24.52 | 22.72 | 21.05 | 22.98 | 31.11 | 53.33 | 33.84 | 37.98 | 33.23 | 20.79 | 18.0 | 28.53 |

Table 4: 3D occupancy prediction performance on the Occ3D-Waymo dataset. Cons. Cone represents the construction cone.

| Method | GO | vehicle | bicyclist | pedestrian | sign | traffic light | pole | Cons. Cone | bicycle | motorcycle | building | vegetation | tree trunk | road | sidewalk | mIoU |
|---|---|---|---|---|---|---|---|---|---|---|---|---|---|---|---|---|
| BEVDet [14] | 0.13 | 13.06 | 2.17 | 10.15 | 7.80 | 5.85 | 4.62 | 0.94 | 1.49 | 0.0 | 7.27 | 10.06 | 2.35 | 48.15 | 34.12 | 9.88 |
| TPVFormer [16] | 3.89 | 17.86 | 12.03 | 5.67 | 13.64 | 8.49 | 8.90 | 9.95 | 14.79 | 0.32 | 13.82 | 11.44 | 5.8 | 73.3 | 51.49 | 16.76 |
| BEVFormer [22] | 3.48 | 17.18 | 13.87 | 5.9 | 13.84 | 2.7 | 9.82 | 12.2 | 13.99 | 0.0 | 13.38 | 11.66 | 6.73 | 74.97 | 51.61 | 16.76 |
| **CTF-Occ (Ours)** | 6.26 | 28.09 | 14.66 | 8.22 | 15.44 | 10.53 | 11.78 | 13.62 | 16.45 | 0.65 | 18.63 | 17.3 | 8.29 | 67.99 | 42.98 | 18.73 |
| LiDAR-Only | 1.01 | 57.41 | 35.31 | 20.33 | 11.7 | 13.01 | 36.21 | 7.81 | 0.13 | 0.0 | 57.83 | 54.71 | 27.07 | 69.15 | 54.47 | 29.74 |
| BEVFormer-Fusion | 5.11 | 64.61 | 52.35 | 21.52 | 32.74 | 17.1 | 42.62 | 27.75 | 13.36 | 0.05 | 63.65 | 60.51 | 35.64 | 81.89 | 66.84 | 39.05 |

**Architecture.** We extend two main-stream BEV models – BEVDet [14] and BEVFormer [22] to the 3D occupancy prediction task. We replace their original detection decoders with the occupancy decoder adopted in our CTF-Occ network and remain their BEV feature encoders. We employ ResNet-101 [12] pretrained on FCOS3D [46] as the image backbone and the image size is resized to $(640 \times 960)$ for Occ3D-Waymo and $(928 \times 1,600)$ for Occ3D-nuScenes. We also evaluate three existing 3D occupancy prediction methods – MonoScene [5], TPVFormer [16], and OccFormer [53] on our proposed Occ3D datasets. Additionally, we conduct experiments using LiDAR as an input on the Waymo dataset. "LiDAR-Onl" refers to adopting single frame LiDAR as input. Voxelization is applied with a voxel size of [0.1, 0.1, 0.4] on the x, y, and z axes respectively. Subsequently, a ResNet is employed to extract dense voxel features, which are then fed to the occupancy prediction head. The "BEVFormer-Fusio" method incorporates both camera and LiDAR inputs. We extract features from the same LiDAR branch and fuse them with the camera features captured by BEVFormer in the BEV space.

Our proposed CTF-Occ adopts a learnable voxel embedding with a shape of $200 \times 200 \times 256$. The voxel embedding will first pass through four encoder layers without token selection. There are three pyramid stage levels for the Occ3D-Waymo dataset, and the resolution of the z-axis in each stage is 8, 16, and 32. The resolution of the z-axis in each stage for the Occ3D-nuScenes dataset is 8 and 16 for the two pyramid stages. Each stage contains one SCA layer and an incremental token selection module to choose K non-empty voxels with the highest scores. The top-k ratio for the incremental token selection strategy is set to 0.2 for all pyramid stages.

**Loss function.** To optimize the occupancy prediction, we use the OHEM loss for model training $\mathcal{L}_{occ} = \sum_k W_k \mathcal{L}(g_k, p_k)$, where $W_k$, $g_k$, and $p_k$ represent the loss weight, the label, and the prediction result for the $k$-th semantic class. In addition, we supervise the binary classification head in each pyramid level with binary voxel masks. The binary voxel masks are generated by processing the semantic occupancy label at each spatial resolution $s_i$ using $f(g, s_i)$, and the output of the binary classification head in the $i$-th level is denoted as $p_i$. The loss for the binary classification is defined as $\mathcal{L}_{bin} = \sum_i \mathcal{L}(f(g, s_i), p_i)$, where $i$ represents the $i$-th pyramid level.

## 6.2 Comparing with previous methods

**Occ3D-nuScenes.** Table 3 shows the performance of 3D occupancy prediction compared to related methods on the Occ3D-nuScenes dataset. It can be observed that our method performs better in all classes than previous baseline methods under the IoU metric. Our CTF-Occ surpass BEVFormer by 1.65 mIoU. The observations are consistent with those in the Occ3D-Waymo dataset.

**Occ3D-Waymo.** We compare the performance of our CTF-Occ network with state-of-the-art models on our newly proposed Occ3D-Waymo dataset. Results are shown in Table 4. Our method outperforms previous methods by remarkable margins, increasing the mIoU by 1.97. Especially for some objects such as traffic cone and vehicle, our method surpasses the baseline method by 2.88 and 10.23 IoU respectively. This is because we capture the features in the 3D voxel space without compressing the heigh, which will preserve the detailed geometry of objects. The results indicate the effectiveness of our coarse-to-fine voxel encoder.

## 6.3 Ablation study

In this section, we ablate the choices of incremental token selection and OHEM loss. Table 5 shows the results. CC represents traffic cones and PED represents pedestrians. We focus on CC and PED to verify the effectiveness of our implementation on small objects. Both techniques improve performance. Using OHEM loss and top-k token selection produces the best performance. Without the OHEM loss, we only get 14.06 mIoU. Combining the OHEM loss with a random token selection strategy achieves 16.62 mIoU. Using an uncertain token selection strategy with OHEM loss achieve 17.37 mIoU. For token selection, uncertain selection and top-k selection are on par and they significantly outperform the random selection as expected.

Table 5: Ablation study on our model components, performed on the Occ3D-Waymo dataset.

| OHEM Loss | Token Selection Strategy | | | IoU | | mIoU |
|---|---|---|---|---|---|---|
| | random | uncertain | top-k | PED | CC | |
| | | | ✓ | 4.16 | 10.03 | 14.06 |
| ✓ | ✓ | | | 5.07 | 12.95 | 16.62 |
| ✓ | | ✓ | | 6.27 | 13.85 | 17.37 |
| ✓ | | | ✓ | 7.04 | 14.16 | 18.43 |

## 7 Conclusion

We present Occ3D, a large-scale high-quality 3D occupancy prediction benchmark for visual perception. Meanwhile, we present a rigorous label generation protocol and a new model CTF-Occ network for the 3D occupancy prediction task. They are publicly released to facilitate future research.

**Limitations.** Although we meticulously design the dataset generation pipeline to significantly enhance its quality, there are several ways to achieve further improvement:

   i. Sensor Calibration Error: Since we use LiDAR scans to construct high-quality occupancy labels for camera perception, the calibration between LiDAR and cameras becomes critical. Conducting multi-frame aggregation also relies on precise sensor calibration.

  ii. Dynamic and Deformable Objects: For dynamic objects, we extract the points located within the box and aggregate them. However, some dynamic objects may not have box annotations, such as running animals, and some objects may not satisfy the rigid body assumption, like a person swinging their arms. There will be motion blur problems in these cases.

 iii. General Objects: Both the nuScenes and Waymo datasets only annotate limited categories. Out-of-vocabulary objects such as trash cans and traffic cones are all regarded as general objects. Further human annotation to provide fine-grained details will help in reproducing an intelligence with unbounded understanding and benefit auto-driving research.

**Acknowledgments.** This work is supported by the National Key R&D Program of China (2022ZD0161700).

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
