# Appendix

## A  Occ3D Dataset

We publish the Occ3D dataset, benchmark, develop kit, data format and annotation instructions at our website Page-Occ3D. It is our priority to protect the privacy of third parties. We bear all responsibility in case of violation of rights, etc., and confirmation of the data license.

**Terms of use, privacy and License.** The Occ3D-nuScenes and Occ3D-Waymo dataset is published under MIT license, which means everyone can use this dataset for non-commercial research purpose. The original nuScenes dataset is released under the CC BY-NC-SA 4.0. The original Waymo dataset is released under the Waymo Dataset License Agreement for Non-Commercial Use (August 2019) License.

**Data maintenance.** Data is stored in Google Drive for global users, and the Occ3D-nuScenes is stored in here and the link for Occ3D-Waymo is stored in here. We will maintain the data for a long time and check the data accessibility on a regular basis.

**Benchmark and code.** Benchmark-Occ3D-nuScenes provides benchmark results of Occ3D-nuScenes. The label generation code will be released upon acceptance.

**Data statistics.** For Occ3D-Waymo, there are 798 scenes for training, 202 scenes for valuation, 150 scenes for testing, and 200,000 frames in total. For Occ3D-nuScenes, there are 700 scenes for training, 150 scenes for valuation, 150 scenes for testing, and 40,000 frames in total.

**Limitations.** The proposed label generation pipeline does not achieve perfect reconstruction and is limited in several ways: it relies on precise sensor calibration, it does not handle deformable objects, etc. Future work will aim to address these issues.

## B  Mesh Reconstruction

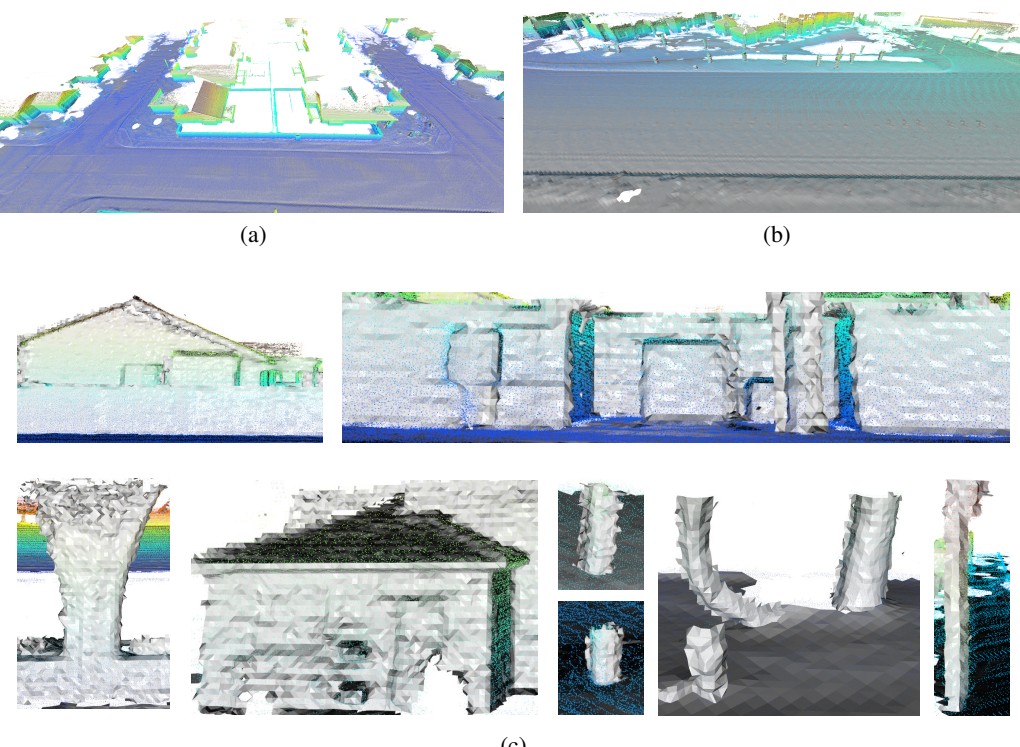

(a)

(b)

(c)

Figure 1: **Mesh reconstruction visualization.** (a) and (b): A couple of scenes after mesh reconstruction. Blue points are the aggregated points, and the gray surface is the reconstructed mesh. (c): Some reconstructed objects, including houses, walls, trees, fire hydrants, and poles.

We apply mesh reconstruction on the aggregated point cloud, and then resample to create a denser voxel representation. In Figures 1a and 1b, the color points represent the aggregated point cloud. It is evident that there are still holes between the points in the original point cloud, which, if converted directly to voxels, would result in many holes with a small voxelization size. After the mesh reconstruction, not only are these holes eliminated, but noisy areas are also effectively smoothed out. Figure 1c shows the results of mesh reconstruction on some objects, including houses, walls, trees, fire hydrants, and poles. As can be observed from the figure, mesh reconstruction is able to effectively perform high-quality surface reconstruction on areas of the objects where point clouds are present.

## C   General Objects

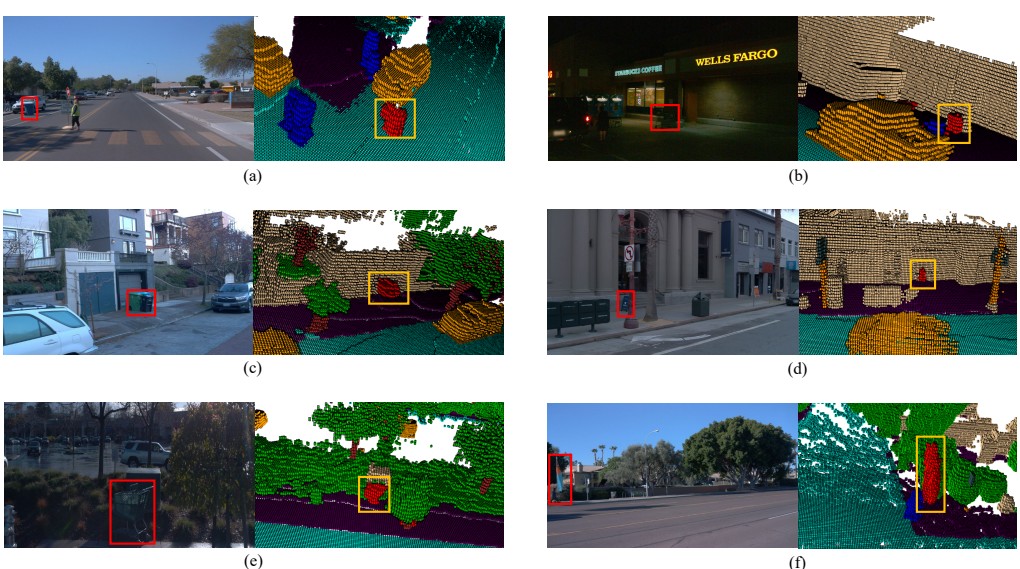

Figure 2: **General objects in our Occ3D benchmark.** We mark the general objects with red boxes in the camera view and yellow boxes in the voxel view.

One of the key advantages of the 3D semantic occupancy prediction task is the potential to handle General Objects (**GOs**), or unknown objects. Different from 3D object detection which pre-defines categories of all the objects, 3D occupancy prediction handles arbitrary objects with occupancy grids and semantics. The geometries of objects are generally represented by voxels including out-of-vocabulary objects labeled as ("occupied", "unknown"). This ability to represent and detect general objects makes the task more general and suitable for autonomous driving perception. Thus, we present a method using the clustering algorithm to handle "unknown" objects.

We showcase several examples of GOs in our Occ3D benchmark in Figure 2. Figure 2(a) and (c) depict a dustbin, while Figure 2(b) and (e) show a shopping cart. Figure 2(d) displays a board on the sidewalk. Figure 2(f) features a flying banner. In each case, the voxels within the bounding box represent the corresponding GO.

## D   Visibility

**Ray casting.** Both the Aggregated LiDAR and Camera visibility calculation heavily depend on a ray-casting algorithm, which is described in detail in Algorithm 1. The algorithm's execution is divided into two stages: the initialization phase (Lines 4 to 32) and the incremental traversal phase (Lines 33 to 65).

During the initialization phase, several parameters are determined: the ray direction $step$, the starting voxel coordinates $cur\_voxel$, the ending voxel coordinates $last\_voxel$, the first voxel boundary $tMax$, and $tDelta$ which defines the distance traversed along the ray when crossing a voxel. The algorithm initiates at the ray's origin. It traverses each voxel in an interval order and continues looping until it encounters the last voxel within the specified range.

The $EPS$ hyper-parameter, set to $1e-9$, is used to nudge the start and end points of the ray slightly inside the traversed voxels to handle edge cases where a ray exactly intersects a voxel boundary. The $DISTANCE$ hyper-parameter, set to 0.5, determines the traversal threshold for the voxel grid, ensuring the ray stops casting when it exceeds the grid.

---

**Algorithm 1:** Ray Casting

---

**Data:** ray_start $\in$ List[3], ray_end $\in$ List[3], pc_range $\in$ List[6], voxel_size $\in$ List[3], spatial_shape $\in$ List[3]
**Result:** cur_voxel $\in$ List[3]
**Function** *ray_casting*:

    $new\_ray\_start[0:3] \leftarrow ray\_start[0:3] - pc\_range[0:3]$
    $new\_ray\_end[0:3] \leftarrow ray\_end[0:3] - pc\_range[0:3]$
    **for** *k in 0 to 2* **do**
        $ray[k] \leftarrow new\_ray\_end[k] - new\_ray\_start[k]$
        **if** $ray[k] \geq 0$ **then**
            $step[k] \leftarrow 1$
        **else**
            $step[k] \leftarrow -1$
        **if** $ray[k] \neq 0$ **then**
            $tDelta[k] \leftarrow (step[k] * voxel\_size[k])/ray[k]$
        **else**
            $tDelta[k] \leftarrow FLOAT\_MAX$
        **end**
        $new\_ray\_start[k] \leftarrow new\_ray\_start[k] + step[k] * voxel\_size[k] * EPS$
        $new\_ray\_end[k] \leftarrow new\_ray\_end[k] - step[k] * voxel\_size[k] * EPS$
        $cur\_voxel[k] \leftarrow \lfloor new\_ray\_start[k]/voxel\_size[k] \rfloor$
        $last\_voxel[k] \leftarrow \lfloor new\_ray\_end[k]/voxel\_size[k] \rfloor$
    **end**
    **for** *k in 0 to 2* **do**
        **if** $ray[k] \neq 0$ **then**
            $cur\_coordinate \leftarrow cur\_voxel[k] * voxel\_size[k]$
            **if** $step[k] < 0$ **and** $cur\_coordinate < new\_ray\_start[k]$ **then**
                $tMax[k] \leftarrow cur\_coordinate$
            **else**
                $tMax[k] \leftarrow cur\_coordinate + step[k] * voxel\_size[k]$
            **end**
            $tMax[k] \leftarrow (tMax[k] - new\_ray\_start[k])/ray[k]$
        **else**
            $tMax[k] \leftarrow FLOAT\_MAX$
        **end**
    **end**
    **while** $step * (cur\_voxel - last\_voxel) < DISTANCE$ **do**
        /* Determine the axis to move based on tMax comparison */
        **if** *tMax[0] < tMax[1]* **then**
            **if** *tMax[0] < tMax[2]* **then**
                $cur\_voxel[0] \leftarrow cur\_voxel[0] + step[0]$
                **if** *cur_voxel [0] < 0 or cur_voxel [0] $\geq$ spatial_shape[0]* **then**
                    **break**
                **end**
                $tMax[0] \leftarrow tMax[0] + tDelta[0]$
            **else**
                $cur\_voxel[2] \leftarrow cur\_voxel[2] + step[2]$
                **if** *cur_voxel [2] < 0 or cur_voxel [2] $\geq$ spatial_shape[2]* **then**
                    **break**
                **end**
                $tMax[2] \leftarrow tMax[2] + tDelta[2]$
            **end**
        **else**
            **if** *tMax[1] < tMax[2]* **then**
                $cur\_voxel[1] \leftarrow cur\_voxel[1] + step[1]$
                **if** *cur_voxel [1] < 0 or cur_voxel [1] $\geq$ spatial_shape[1]* **then**
                    **break**
                **end**
                $tMax[1] \leftarrow tMax[1] + tDelta[1]$
            **else**
                $cur\_voxel[2] \leftarrow cur\_voxel[2] + step[2]$
                **if** *cur_voxel [2] < 0 or cur_voxel [2] $\geq$ spatial_shape[2]* **then**
                    **break**
                **end**
                $tMax[2] \leftarrow tMax[2] + tDelta[2]$
            **end**
        **end**
        yield $cur\_voxel$
    **end**

---

**Aggregated LiDAR visibility.** The calculation of aggregated LiDAR visibility is described in Algorithm 2. The term $points$ denotes the aggregated point cloud, and $point origin$ stands for the corresponding LiDAR origin. Initially (Line 2), $voxel\_state$ is set to $NOT\_OBSERVED$, and $voxel\_label$ is initialized as $FREE\_LABEL$. In Lines 12-13, for each voxel related to a point, the voxel occupancy counts $voxel\_occ\_count$ is accumulated by one, and the $voxel\_label$ is assigned the label of the current point. For any voxel that the ray passes through, the voxel free count $voxel\_free\_count$ is accumulated. Finally, the state of voxels with $voxel\_free\_count$ greater than zero is set as $FREE$, and those with $voxel\_occ\_count$ greater than zero are set as $OCCUPIED$. Despite a large number of points, often up to 2 million, the computation time is optimized to around 10 milliseconds by utilizing parallel processing on GPU, as shown in Line 5.

**Algorithm 2:** Aggregated LiDAR Visibility

**Data:** $points\_origin \in Tensor(N,3), points \in Tensor(N,3), points\_label \in Tensor(N,), pc\_range \in$
    $List[6], voxel\_size \in List[3], spatial\_shape \in List[3]$
**Result:** $voxel\_state \in Tensor(H,W,Z), voxel\_label \in Tensor(H,W,Z)$
**Function** $calculate\_LiDAR\_visibility$:
    Initialize $voxel\_occ\_count \in Tensor(H,W,Z), voxel\_free\_count \in Tensor(H,W,Z)$
    $voxel\_state \leftarrow NOT\_OBSERVED, voxel\_label \leftarrow FREE\_LABEL, voxel\_occ\_count \leftarrow 0, voxel\_free\_count \leftarrow 0$
    Filter points, $points\_origin$, and $points\_label$ within $pc\_range$
    **for** *i in 0 to N* **do**
        $ray\_start \leftarrow points[i]$
        $ray\_end \leftarrow points\_origin[i]$
        **for** *k in 0 to 2* **do**
            $target\_voxel[k] \leftarrow \lfloor \frac{(ray\_start[k] - pc\_range[k])}{voxel\_size[k]} \rfloor$
        **end**
        **if** *target_voxel $\in$ spatial_shape* **then**
            $atomicAdd(voxel\_occ\_count[target\_voxel], 1)$
            $voxel\_label[target\_voxel] \leftarrow points\_label[i]$
        **for** *voxel_index in ray_casting(ray_start, ray_end, pc_range, voxel_size, spatial_shape)* **do**
            $atomicAdd(voxel\_free\_count[voxel\_index], 1)$
        **end**
    **end**
    $voxel\_state[voxel\_free\_count>0] \leftarrow FREE$
    $voxel\_state[voxel\_occ\_count>0] \leftarrow OCCUPIED$

**Camera visibility.** The calculation of camera visibility is described in Algorithm 3. In Line 3, $update\_voxel\_state$ is initialized to $NOT\_OBSERVED$, and then some voxels marked as $OCCUPIED$ and $FREE$ under the LiDAR view are further assigned to $NOT\_OBSERVED$. For each pixel in each camera image, a virtual point is generated at a significant distance away, as illustrated in Lines 5-9. Then points are transformed from the image coordinates to the ego coordinate system (Line 10). The camera origin serves as the origin for the virtual point, and is similarly transformed to the ego coordinate system in Lines 12-15. In Lines 24-34, the $update\_voxel\_state$ for voxels traversed by the pixel ray is assigned the same value as $voxel\_state$. The $DEPTH\_MAX$ hyper-parameter, which is set to $1e3$, acts as a surrogate for substantial depth. To enhance computational efficiency, each ray's operation executes concurrently on a GPU, as demonstrated in Line 21.

**Algorithm 3:** Camera Visibility

**Data:** $Image \in Tensor(K,h,w), P_{cam} \in Tensor(K,4,4), P_{cam2ego} \in Tensor(K,4,4), P_{ego2global} \in Tensor(K,4,4), P_{intrinsics} \in Tensor(K,$
    4, 4), voxel_state $\in$ Tensor(H,W,Z), voxel_label $\in$ Tensor(H,W,Z), pc_range $\in$ List[6], voxel_size $\in$ List[3], spatial_shape $\in$ List[3]
**Result:** $update\_voxel\_state \in$ Tensor(H,W,Z)
**Function** *calculate_Camera_visibility*:
    Initialize $origins\_list \leftarrow List[], uvs\_list \leftarrow List[]$
    $update\_voxel\_state \leftarrow NOT\_OBSERVED$
    **for** *k in 0 to K* **do**
        /* Generate meshgrid points for image */
        $uvs \in Tensor(2, h*w) \leftarrow meshgrid(Image[k])$
        $depth \leftarrow Full((1, h*w), fill\_value = DEPTH\_MAX)$
        $uvs \leftarrow concatenate([uvs, Ones((1, h*w))])$
        $uvs \leftarrow uvs * depth.repeat(3, 1)$
        Convert uvs from Image to ego coordinate using $P_{cam2ego}[k]$ and $P_{intrinsics}[k]$
        $uvs \leftarrow uvs.transpose()$
        $origin \leftarrow Zeros((4, 4))$
        $origin[3, 3] \leftarrow 1$
        Convert origin from Camera to ego coordinate using $P_{cam2ego}[k]$
        $origin \leftarrow origin.reshape(1, -1).expand(uvs.shape[0], 3);$
        Add $uvs$ to $uvs\_list$
        Add $origin$ to $origins\_list$
    **end**
    $uv2points \leftarrow concatenate(uvs\_list\_list)$
    $origins \leftarrow concatenate(origins\_list)$
    **for** *i in 0 to N* **do**
        $ray\_start \leftarrow origins[i]$
        $ray\_end \leftarrow uv2points[i]$
        **for** *voxel_index in ray_casting(ray_start, ray_end, pc_range, voxel_size, spatial_shape)* **do**
            **if** *voxel_state == OCCUPIED* **then**
                $update\_voxel\_state \leftarrow OCCUPIED$
            **else**
                **if** *voxel_state == FREE* **then**
                    $update\_voxel\_state \leftarrow FREE$
                **else**
                  $update\_voxel\_state \leftarrow NOT\_OBSERVED$
                **end**
            **end**
        **end**
    **end**

**Visualization.** Accurately determining the visibility of a voxel is crucial for the 3D occupancy prediction task, as it helps eliminate training and evaluation ambiguity. As discussed in Section 4, Figure 3 illustrates the "unobserved" voxels in the camera view due to occlusion. The yellow-green cube represents the ego vehicle, and the red-colored voxels are the "unobserved" voxels determined by our visibility mask generation procedure. Figure 3(a) shows the blind spots of ego vehicles and how parked vehicles at the roadside occlude the area behind them. Figure. 3(b) mainly shows that in the current camera views, the drivable surface and the buildings

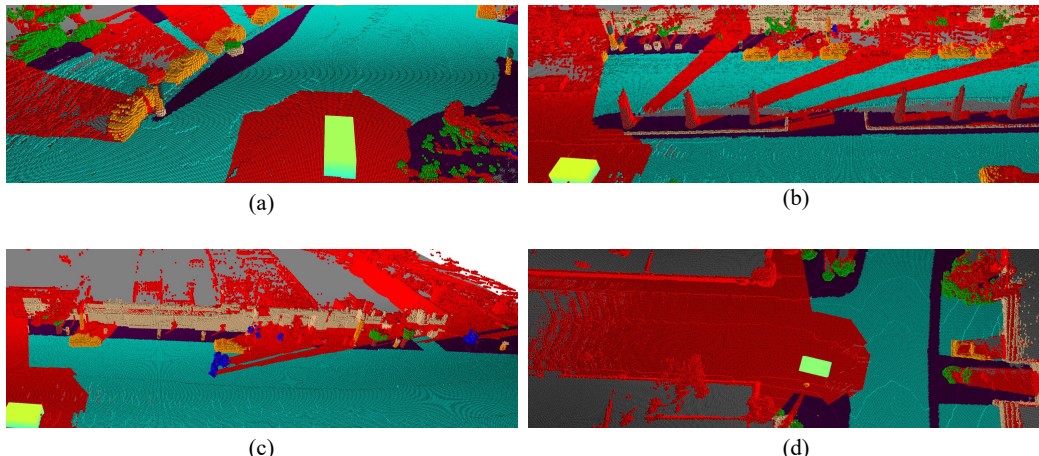

|     |     |
| :-: | :-: |
| (a) | (b) |
| (c) | (d) |

Figure 3: **Occlusion reasoning and camera visibility.** Grey voxels are unobserved in the LiDAR view and red voxels are observed in the accumulative LiDAR view but unobserved in the current camera view.

behind the tree trunks are occluded. In the right part of the image in Figure. 3(c), voxels that represent buildings behind walls are marked as "unobserved". As illustrated in Figure. 3(d), the Waymo dataset doesn't provide the back-view camera image, leading to the blind spots in a certain range of angles behind the vehicle. By accurately determining voxel visibility, we can improve the accuracy and reliability of our 3D occupancy prediction model, which is critical for autonomous driving systems.

## E    3D-2D Consistency

Figure 4 illustrates a visualization of the 3D-2D consistency evaluation conducted using the Waymo dataset. From right to left, the figure displays the original image, the 2D ROI, 3D voxel semantics, and 2D pixel semantics. Vertically, the figure presents the results in the order of CAMERA_FRONT, CAMERA_FRONT_LEFT, CAMERA_LEFT, CAMERA_FRONT_RIGHT, and CAMERA_RIGHT. The result for CAMERA_BACK is notably absent due to the original Waymo dataset not including images from rear-view cameras.

The visualization results demonstrate that the semantic labels for 3D voxels, generated via our auto-labeling method, align consistently with the manually annotated 2D semantic labels. This underscores the effectiveness of our proposed method. In the majority of instances, our proposed 3D-2D consistency calculation method provides an accurate measurement of this consistency. However, in certain situations, such as in Figure 4e where the 2D semantic labels incorrectly annotated a tree trunk as a pole by humans, there can be a notable impact on the 3D-2D consistency metrics.

## F    Datasheet

1. **For what purpose was the dataset created?** Was there a specific task in mind? Was there a specific gap that needed to be filled? Please provide a description.

    - Occ3D was created as a benchmark for 3D Occupancy Prediction task. The goal of this task is to predict the 3D occupancy of the scene. Understanding the 3D surroundings including the background stuffs and foreground objects is important for autonomous driving. In the traditional 3D object detection task, a foreground object is represented by the 3D bounding box. However, the geometrical shape of the object is complex, which can not be represented by a simple 3D box, and the perception of the background stuffs is absent. The benchmark is a voxelized representation of the 3D space, and the occupancy state and semantics of the voxel in 3D space are jointly estimated in this task. The complexity of this task lies in the dense prediction of 3D space given the surround-view images.

2. **Who created the dataset (e.g., which team, research group) and on behalf of which entity (e.g., company, institution, organization)?**

    - This dataset is presented by Tsinghua MARS Lab.

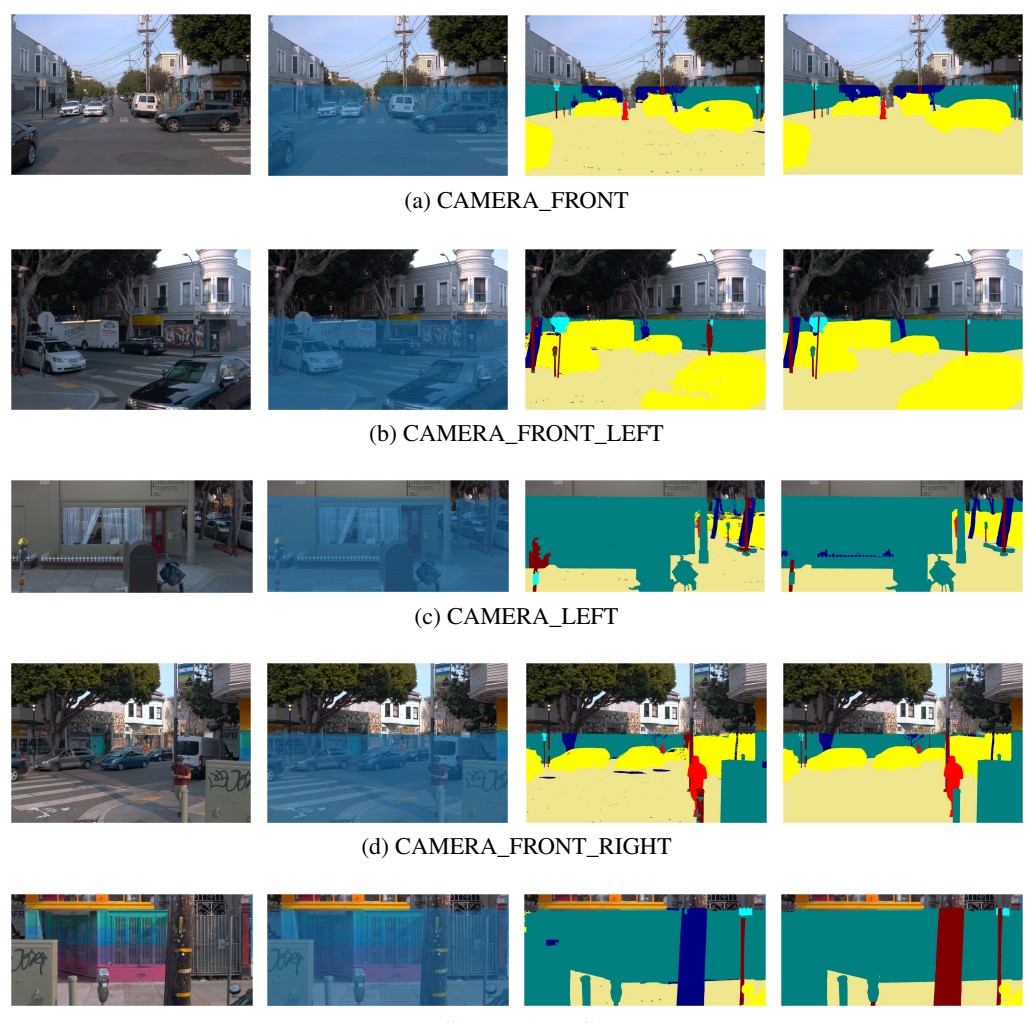

(a) CAMERA_FRONT

(b) CAMERA_FRONT_LEFT

(c) CAMERA_LEFT

(d) CAMERA_FRONT_RIGHT

(e) CAMERA_RIGHT

Figure 4: **Visualization of 3D-2D consistency.** From right to left are the visualization of original images, 2D ROI, 3D voxel semantics, 2D pixel semantics; From top to bottom are the results of CAMERA_FRONT, CAMERA_FRONT_LEFT, CAMERA_LEFT, CAMERA_FRONT_RIGHT, and CAMERA_RIGHT.

3. **Who funded the creation of the dataset?** If there is an associated grant, please provide the name of the grantor and the grant name and number.

   - This work was sponsored by Tsinghua University.

4. **Any other comments?**

   - No.

### F.1 Composition

5. **What do the instances that comprise the dataset represent (e.g., documents, photos, people, countries)?** *Are there multiple types of instances (e.g., movies, users, and ratings; people and interactions between them; nodes and edges)? Please provide a description.*

   - We provide 40,000 samples for Occ3D-nuScenes and 200,000 samples for Occ3D-Waymo. Each sample in Occ3D-nuScenes consists of the following: 6 RGB images; 1 LiDAR point cloud; 1 3D voxel semantic ground-truth; 1 LiDAR visibility mask; 1 camera visibility mask; 1 metadata. Each sample in Occ3D-Waymo consists of the following: 5 RGB images; 1 LiDAR point cloud; 1 3D voxel semantic ground-truth; 1 LiDAR visibility mask; 1 camera

visibility mask; 1 metadata. We made our benchmark openly available on the Occ3D github page(`https://github.com/Tsinghua-MARS-Lab/Occ3D`).

6. **How many instances are there in total (of each type, if appropriate)?**

   - For Occ3D-nuScenes, there are 600 scenes for training, 150 scenes for valuation, 250 scenes for testing, 40,000 frames in total. For Occ3D-Waymo, there are 798 scenes for training, 202 scenes for valuation, 150 scenes for testing, 200,000 frames in total.

7. **Does the dataset contain all possible instances or is it a sample (not necessarily random) of instances from a larger set?** *If the dataset is a sample, then what is the larger set? Is the sample representative of the larger set (e.g., geographic coverage)? If so, please describe how this representativeness was validated/verified. If it is not representative of the larger set, please describe why not (e.g., to cover a more diverse range of instances, because instances were withheld or unavailable).*

   - Both nuScenes and Waymo are open-source datasets. We use the proposed auto-labeling method to derive Occ3D-nuScenes and Occ3D-Waymo. For Occ3D-nuScene, we use the annotated frames(2Hz) in nuScenes, which is representative; For Occ3D-Waymo, we use all samples of Waymo Open dataset.

8. **What data does each instance consist of?** *"Raw" data (e.g., unprocessed text or images) or features? In either case, please provide a description.*

   - Each instance consist of RGB images, LiDAR point cloud, 3D voxel semantic ground-truth, LiDAR visibility mask, camera visibility mask and metadata.

9. **Is there a label or target associated with each instance?** *If so, please provide a description.*

   - There is a 3D voxel semantics label for each instance, which describe the semantic label of each voxel in the 3D scene.

10. **Is any information missing from individual instances?** *If so, please provide a description, explaining why this information is missing (e.g., because it was unavailable). This does not include intentionally removed information, but might include, e.g., redacted text.*

    - No.

11. **Are relationships between individual instances made explicit (e.g., users' movie ratings, social network links)?** *If so, please describe how these relationships are made explicit.*

    - No.

12. **Are there recommended data splits (e.g., training, development/validation, testing)?** *If so, please provide a description of these splits, explaining the rationale behind them.*

    - We use the original data splits in nuScenes and Waymo for Occ3D. For Occ3D-nuScenes, there are 600 train sequences, 150 validation sequences and 200 test sequences; For Occ3D-Waymo, there are 798 train sequences, 202 validation sequences and 150 test squences.

13. **Are there any errors, sources of noise, or redundancies in the dataset?** *If so, please provide a description.*

    - There exist noises in the dataset due to the LiDAR nosies and pose inaccuracies.

14. **Is the dataset self-contained, or does it link to or otherwise rely on external resources (e.g., websites, tweets, other datasets)?** *If it links to or relies on external resources, a) are there guarantees that they will exist, and remain constant, over time; b) are there official archival versions of the complete dataset (i.e., including the external resources as they existed at the time the dataset was created); c) are there any restrictions (e.g., licenses, fees) associated with any of the external resources that might apply to a future user? Please provide descriptions of all external resources and any restrictions associated with them, as well as links or other access points, as appropriate.*

    - We release the Occ3D dataset on our GitHub repository: `https://github.com/Tsinghua-MARS-Lab/Occ3D`. More specifically, please use the following links to visit the documentations and download instructions: Occ3D-Webpage. Our dataset is developed based on existing automonous driving dataset nuScenes and Waymo

15. **Does the dataset contain data that might be considered confidential (e.g., data that is protected by legal privilege or by doctor–patient confidentiality, data that includes the content of individuals' non-public communications)?** *If so, please provide a description.*

    - Our dataset is developed based on nuScenes(developed by Motional )and Waymo (developed by Waymo ), which has already removed confidential data.

16. **Does the dataset contain data that, if viewed directly, might be offensive, insulting, threatening, or might otherwise cause anxiety?** *If so, please describe why.*

    • No.

17. **Does the dataset relate to people?** *If not, you may skip the remaining questions in this section.*

    • No.

18. **Does the dataset identify any subpopulations (e.g., by age, gender)?**

    • No.

19. **Is it possible to identify individuals (i.e., one or more natural persons), either directly or indirectly (i.e., in combination with other data) from the dataset?** *If so, please describe how.*

    • No.

20. **Does the dataset contain data that might be considered sensitive in any way (e.g., data that reveals racial or ethnic origins, sexual orientations, religious beliefs, political opinions or union memberships, or locations; financial or health data; biometric or genetic data; forms of government identification, such as social security numbers; criminal history)?** *If so, please provide a description.*

    • No.

21. **Any other comments?**

    • No.

### F.2    Collection Process

22. **How was the data associated with each instance acquired?** *Was the data directly observable (e.g., raw text, movie ratings), reported by subjects (e.g., survey responses), or indirectly inferred/derived from other data (e.g., part-of-speech tags, model-based guesses for age or language)? If data was reported by subjects or indirectly inferred/derived from other data, was the data validated/verified? If so, please describe how.*

    • Our data is developing based on published data nuScenes and Waymo using a designed auto-labeling method mentioned before.

23. **What mechanisms or procedures were used to collect the data (e.g., hardware apparatus or sensor, manual human curation, software program, software API)?** *How were these mechanisms or procedures validated?*

    • We ran a auto-labeling script in python to generate the ground-truth labels. We use hundred of small CPU nodes, and few GPU nodes. They were validated by manual inspection of the results and 2D-3D consistency quality check we described in the body part.

24. **If the dataset is a sample from a larger set, what was the sampling strategy (e.g., deterministic, probabilistic with specific sampling probabilities)?**

    • We use full-set provided by nuScenes and Waymo.

25. **Who was involved in the data collection process (e.g., students, crowdworkers, contractors) and how were they compensated (e.g., how much were crowdworkers paid)?**

    • No crowdworkers were involved in the curation of the dataset. Open-source researchers and developers enabled its creation for no payment.

26. **Over what timeframe was the data collected? Does this timeframe match the creation timeframe of the data associated with the instances (e.g., recent crawl of old news articles)?** *If not, please describe the timeframe in which the data associated with the instances was created.*

    • The 3D occupancy ground-truth data was generated in 2023, while the source sensor data was created in 2019 for nuScenes and 2020 for Waymo.

27. **Were any ethical review processes conducted (e.g., by an institutional review board)?** *If so, please provide a description of these review processes, including the outcomes, as well as a link or other access point to any supporting documentation.*

    • The source sensor data for nuScenes and Waymo had been conducted ethical review processes by Motional and Waymo, which can be referred to nuScenes and Waymo, respectively.

28. **Did you collect the data from the individuals in question directly, or obtain it via third parties or other sources (e.g., websites)?**

    • We retrieve the data from the open source datasets nuScenes and Waymo.

29. **Were the individuals in question notified about the data collection?** *If so, please describe (or show with screenshots or other information) how notice was provided, and provide a link or other access point to, or otherwise reproduce, the exact language of the notification itself.*

- The Occ3D dataset is developed based on open-source dataset and following the open-source license.

30. **Did the individuals in question consent to the collection and use of their data?** *If so, please describe (or show with screenshots or other information) how consent was requested and provided, and provide a link or other access point to, or otherwise reproduce, the exact language to which the individuals consented.*

- The Occ3D dataset is developed on open-source dataset and obey the license.

31. **If consent was obtained, were the consenting individuals provided with a mechanism to revoke their consent in the future or for certain uses?** *If so, please provide a description, as well as a link or other access point to the mechanism (if appropriate).*

- Users have a possibility to check for the presence of the links in our dataset leading to their data on public internet by using the search tool provided by Occ3D, accessible at Occ3D-Webpage. If users wish to revoke their consent after finding sensitive data, they can contact the hosting party and request to delete the content from the underlying website. Please leave the message in GitHub Issue to request removal of the links from the dataset.

32. **Has an analysis of the potential impact of the dataset and its use on data subjects (e.g., a data protection impact analysis) been conducted?** *If so, please provide a description of this analysis, including the outcomes, as well as a link or other access point to any supporting documentation.*

- We develop our dataset based on open source dataset nuScenes and Waymo publised by Motional and Waymo. The published dataset has been seriously considered of it's potential impact and its use on data subjects.

33. **Any other comments?**

- No.

### F.3 Preprocessing, Cleaning, and/or Labeling

34. **Was any preprocessing/cleaning/labeling of the data done (e.g., discretization or bucketing, tokenization, part-of-speech tagging, SIFT feature extraction, removal of instances, processing of missing values)?** *If so, please provide a description. If not, you may skip the remainder of the questions in this section.*

- We use an auto-labeling preprocessing script to generate the 3D voxel semantic labels of the dataset. Beside this, no preprocessing or labelling is done.

35. **Was the "raw" data saved in addition to the preprocessed/cleaned/labeled data (e.g., to support unanticipated future uses)?** *If so, please provide a link or other access point to the "raw" data.*

- Yes, we provide the original open source dataset and the auto-labeled Occ3D dataset.

36. **Is the software used to preprocess/clean/label the instances available?** *If so, please provide a link or other access point.*

- No.

37. **Any other comments?**

- No.

### F.4 Uses

38. **Has the dataset been used for any tasks already?** *If so, please provide a description.*

- No.

39. **Is there a repository that links to any or all papers or systems that use the dataset?** *If so, please provide a link or other access point.*

- No.

40. **What (other) tasks could the dataset be used for?**

- We encourage future researchers to curate Occ3D for several tasks. For instance, we hope that researchers can use the Occ3D we provide to study how to better promote some downstream tasks such as autonomous driving prediction and planning.

41. **Is there anything about the composition of the dataset or the way it was collected and prepro-cessed/cleaned/labeled that might impact future uses?** *For example, is there anything that a future user might need to know to avoid uses that could result in unfair treatment of individuals or groups (e.g., stereotyping, quality of service issues) or other undesirable harms (e.g., financial harms, legal risks) If so, please provide a description. Is there anything a future user could do to mitigate these undesirable harms?*

    - No.

42. **Are there tasks for which the dataset should not be used?** *If so, please provide a description.*

    - Due to the known biases of the dataset, under no circumstance should any models be put into production using the dataset as is. It is neither safe nor responsible. As it stands, the dataset should be solely used for research purposes in its uncurated state.

43. **Any other comments?**

    - No.

### F.5 Distribution

44. **Will the dataset be distributed to third parties outside of the entity (e.g., company, institution, organization) on behalf of which the dataset was created?** *If so, please provide a description.*

    - Yes, the dataset will be open-source.

45. **How will the dataset be distributed (e.g., tarball on website, API, GitHub)?** *Does the dataset have a digital object identifier (DOI)?*

    - The data is available through `https://github.com/Tsinghua-MARS-Lab/Occ3D`.

46. **When will the dataset be distributed?**

    - 31/03/2023 and onward.

47. **Will the dataset be distributed under a copyright or other intellectual property (IP) license, and/or under applicable terms of use (ToU)?** *If so, please describe this license and/or ToU, and provide a link or other access point to, or otherwise reproduce, any relevant licensing terms or ToU, as well as any fees associated with these restrictions.*

    - The Occ3D dataset is published under MIT license, which means everyone can use this dataset for non-commercial research purpose. The original nuScenes dataset is released under the CC BY-NC-SA 4.0. The original Waymo dataset is released under the Waymo Dataset License Agreement for Non-Commercial Use (August 2019) License.

48. **Have any third parties imposed IP-based or other restrictions on the data associated with the instances?** *If so, please describe these restrictions, and provide a link or other access point to, or otherwise reproduce, any relevant licensing terms, as well as any fees associated with these restrictions.*

    - The original nuScenes dataset is released under the CC BY-NC-SA 4.0, and the for the restrictions, please refer to nuScenes. The original Waymo dataset is released under the Waymo Dataset License Agreement for Non-Commercial Use (August 2019) License, and the for the restrictions, please refer to Waymo.

49. **Do any export controls or other regulatory restrictions apply to the dataset or to individual instances?** *If so, please describe these restrictions, and provide a link or other access point to, or otherwise reproduce, any supporting documentation.*

    - No.

50. **Any other comments?**

    - No.

### F.6 Maintenance

51. **Who will be supporting/hosting/maintaining the dataset?**

    - Tsinghua MARS Lab will support hosting of the dataset.

52. **How can the owner/curator/manager of the dataset be contacted (e.g., email address)?**

    - `https://github.com/Tsinghua-MARS-Lab/Occ3D/issues`

53. **Is there an erratum?** *If so, please provide a link or other access point.*

- There is no erratum for our initial release. Errata will be documented as future releases on the dataset website.

54. **Will the dataset be updated (e.g., to correct labeling errors, add new instances, delete instances)?** *If so, please describe how often, by whom, and how updates will be communicated to users (e.g., mailing list, GitHub)?*

    - We will continue to support Occ3D dataset.

55. **If the dataset relates to people, are there applicable limits on the retention of the data associated with the instances (e.g., were individuals in question told that their data would be retained for a fixed period of time and then deleted)?** *If so, please describe these limits and explain how they will be enforced.*

    - No.

56. **Will older versions of the dataset continue to be supported/hosted/maintained?** *If so, please describe how. If not, please describe how its obsolescence will be communicated to users.*

    - Yes. We will continue to support Occ3D dataset in our github page.

57. **If others want to extend/augment/build on/contribute to the dataset, is there a mechanism for them to do so?** *If so, please provide a description. Will these contributions be validated/verified? If so, please describe how. If not, why not? Is there a process for communicating/distributing these contributions to other users? If so, please provide a description.*

    - Yes, they can driectly developing on open scource dataset nuScenes and Waymo dataset or concat us via GitHub Issue.

58. **Any other comments?**

    - No.