# OpenReview forum: "Occ3D: A Large-Scale 3D Occupancy Prediction Benchmark for Autonomous Driving"
_NeurIPS.cc/2023/Track/Datasets_and_Benchmarks — NeurIPS 2023 Datasets and Benchmarks Poster_

### Official Review · Reviewer_S7Q7 · 2023-07-04

**Rating:** 5
**Confidence:** 5
**Clarity:** Yes, this paper is written clearly, a…

**Strengths:**

1. The motivation is clear. The paper addresses an important issue, that 3D bounding box has some limitations in depicting the real 3D scene. Therefore, the author proposes Occ3D to facilitate the 3D occupancy prediction task, which is a better representation in handling general objects and geometric details.

2. The author constructs Occ3D on two prevailing autonomous driving datasets, the Waymo and the nuScenes, adding diversity of the benchmark in the community.

3. The author takes a quality check on the proposed dataset by using 3D-2D consistency, which validates its effectiveness.

**Additional Feedback:**

My full concern is listed above. Please respond accordingly.

**Correctness:**

The claims made in the submission sounds correct to me. But I remain some concern on the construction and the evaluation of the Occ3D-nuScenes. Please see the opportunity and limitation part.

**Documentation:**

I may need more detail about the data construction part as stated in limitations part. Also please check about the license issue of Waymo and nuScenes, I am not sure post-processing their data to release a new benchmark is ok or not.

**Limitations:**

1. A major question is: why the benchmarked methods are all only taking camera images as the input? My understanding is that methods with lidar or radar signals can also be evaluated by the proposed dataset. Please correct me if my understanding is inaccurate.

2. nuScenes dataset lacks z-axis translation in their map/imu metadata, which will lead to the misalignment of point clouds during their aggregation across entire scenes. This, in turn, leads to suboptimal quality of the labels. It appears that the paper seems to have neglected to account for this particular issue.

3. About category in Occ3D nuscenes. Is it a one-to-one mapping between Occ3D-nuScenes and panoptic nuScenes? It seems like Occ3D maps the “ignore” class in panoptic nuScenes to “others” in Occ3D. If so, what is the motivation of adding this class into evaluation? This class indicates noise point, is very small in number, usually for auxiliary training, and is not taken into account in the semantic segmentation task of nuScenes.

**Opportunities For Improvement:**

1. Missing some related work on method part. Some camera-only 3D occupancy prediction methods such as VoxFormer, OccDepth, BEV-IO and so on. Please consider revising accordingly. Notably that the LiDAR segmentation ground truth comes from the Panoptic nuScenes[1].

2. In nuScenes there are two coordinate systems, one is lidar and the other is ego-frame. Which one does the author use for the 3D occupancy ground truth?

[1] Whye Kit Fong, Rohit Mohan, Juana Valeria Hurtado, Lubing Zhou, Holger Caesar, Oscar Beijbom, and Abhinav Valada. Panoptic nuscenes: A large-scale benchmark for lidar panoptic segmentation and tracking. In ICRA, 2022.

**Relation To Prior Work:**

The paper misses some citation to the prior work, please see in opportunity part.

**Summary And Contributions:**

The limitations of 3D bounding box (1. erases the geometric details of objects; 2. uncommon categories are often ignored and not labeled) call for a general and coherent representation that can model the detailed geometry.

This paper introduces Occ3D, a high-quality 3D occupancy prediction benchmark to facilitate research in this emerging area; they design a rigorous automatic label generation pipeline for constructing the Occ3D benchmark, with comprehensive validation of the effectiveness of the pipeline; with Occ3D, they benchmark existing models and propose a new CTF-Occ network that achieves superior performance.

---

> ### Author Response · Authors · 2023-08-25
> **Response to Reviewer S7Q7**
>
> We appreciate the reviewer's comment and address each of the concerns separately below.
>
> **1. Missing some related work**
> *Original comment:
> Missing some related work on method part.*
>
> ***Response:***
> Thanks for pointing this. We have updated the related work in the revised version of our paper.
>
> **2. Coordinate system**
> *Original comment:
> Which one does the author use for the 3D occupancy ground truth?*
>
> ***Response:***
> We use the ego-frame coordinate system for both the Occ3D-nuScenes and Occ3D-Waymo datasets.
>
> **3. Why only evaluate vision-centric methods**
> *Original comment:
> A major question is: why the benchmarked methods are all only taking camera images as the input?*
>
> ***Response:***
> We primarily focus on vision-centric perception due to its low-cost and rich semantic content. 3D perception from visual information is a foundational and long-standing challenge, the dataset was proposed to benefit the research community focused on this topic.
>
> We also conducted experiments using LiDAR as an input on the Waymo dataset, adhering to the settings detailed in Section 6.1 of our paper. "LiDAR-Only" use a single frame LiDAR as input. We applied voxelization with a voxel size of [0.1, 0.1, 0.4] on the x, y, and z axes respectively. Then, a ResNet is employed to extract dense voxel features, which are subsequently fed to the same occupancy prediction head described in our paper. The "BEVFormer-Fusion" method incorporates both camera and LiDAR inputs. We extract features from the same LiDAR branch and fuse them with the camera features captured by BEVFormer in the BEV space. The code is aviable in our code repository.
> | method | GO | vehicle | bicyclist | pedestrian | sign | traffic light | pole | C.C | bicycle | motorcycle | building | vegetation | tree trunk | road | sidewalk | mIoU |
> | ---- | ---- | ---- | ---- | ---- | ---- | ---- | ---- | ---- | ---- | ---- | ---- | ---- | ---- | ---- | ---- | ---- |
> BEVFormer-camera | 3.48 | 17.18 | 13.87 | 5.9 | 13.84 | 2.7 | 9.82 | 12.2 | 13.99 | 0.0 | 13.38 | 11.66 | 6.73 | 74.97 | 51.61 | 16.76 |
> LiDAR-Only | 1.01 | 57.41 | 35.31 | 20.33 | 11.7 | 13.01 | 36.21 | 7.81 | 0.13 | 0.0 | 57.83 | 54.71 | 27.07 | 69.15 | 54.47 | 29.74 |
> BEVFormer-Fusion | 5.11 | 64.61 | 52.35 | 21.52 | 32.74 | 17.1 | 42.62 | 27.75 | 13.36 | 0.05 | 63.65 | 60.51 | 35.64 | 81.89 | 66.84 | 39.05 |
>
> **4. nuScenes dataset lacks z-axis translation**
> *Original comment:
> nuScenes dataset lacks z-axis translation in their map/imu metadata, which will lead to the misalignment of point clouds during their aggregation across entire scenes.*
>
> ***Response:***
> The absence of z-axis translation in the nuScene dataset leads to misalignment. To enhance dataset quality, we employ additional optimizations, including ICP and ground fitting. Furthermore, we adopt a relatively large voxel size of 0.4m for the final label, which makes the misalignment negligible. We also observed that the pose is notably accurate in the Waymo dataset; hence, we establish occ3d-Waymo to enhance diversity.
>
>
> **5. Category in Occ3D nuscenes**
> *Original comment:
> Is it a one-to-one mapping between Occ3D-nuScenes and panoptic nuScenes? It seems like Occ3D maps the “ignore” class in panoptic nuScenes to “others” in Occ3D. If so, what is the motivation of adding this class into evaluation?*
>
> **Response:***
> It is not a direct one-to-one mapping between Occ3D-nuScenes and panoptic nuScenes. The "ignore" class in panoptic nuScenes includes several sub-categories, as detailed in the table below:
> | lidar segmentation index| lidar segmentation class | nuScenes-lidarseg general class |
> | ------ | ------ | ------ |
> | 0 | ignore | animal |
> | 0 | ignore | human.pedestrian.personal_mobility |
> | 0 | ignore | human.pedestrian.stroller |
> | 0 | ignore | human.pedestrian.wheelchair |
> | 0 | ignore | movable_object.debris |
> | 0 | ignore | movable_object.pushable_pullable |
> | 0 | ignore | static_object.bicycle_rack |
> | 0 | ignore | vehicle.emergency.ambulance |
> | 0 | ignore | vehicle.emergency.police |
> | 0 | ignore | noise |
> | 0 | ignore | static.other |
> | 0 | ignore | vehicle.ego |
>
> The nuScenes-lidarseg comprises 32 detailed categories, as described in the nuScenes repository[https://github.com/nutonomy/nuscenes-devkit/blob/master/python-sdk/nuscenes/lidarseg/class_histogram.py]. When constructing our Occ3D-nuScenes, we exclude the "noise" and "vehicle.ego" categories. The classes remaining, as shown in the preceding table, are labeled as General Objects (GOs). While these GOs are rare, it's imperative to include them in evaluations. This is crucial for perception tasks prioritizing with safety considerations since GOs are typically undetected by 3D object detection with predefined categories.
>
> **6. license issue**
> *Original comment:
> I am not sure post-processing their data to release a new benchmark is ok or not.*
>
> ***Response:***
> Post-processing the nuScene and Waymo data to release a new benchmark is allowed.

---

> ### Author Response · Authors · 2023-08-28
>
> Thank you for reviewing our paper and providing thoughtful comments on this paper! Since you reduce the score from 5 to 4, we appreciate any further comments and strive to address your concerns. If there are particular details you think we have omitted that you would like to review or discuss, it would be our pleasure to post them here for your consideration.

---

> > ### Comment · Reviewer_S7Q7 · 2023-08-30
> >
> > Thanks for the response. I still not 100% convinced on two issues:
> >
> > 1. nuScenes dataset lacks z-axis translation
> >
> > 1.1 "To enhance dataset quality, we employ additional optimizations, including ICP and ground fitting."
> > By doing these post-processing, is the occupancy ground truth better? If so, is there any stats / visualization to support this? I feel like it is a hard issue to be solved and it poses an apparent shortcoming in the construction of ground truth.
> >
> > 1.2 "Furthermore, we adopt a relatively large voxel size of 0.4m for the final label, which makes the misalignment negligible."
> > By using large voxel size, isn't it whittling away the advantage of using occupancy? I feel like there should be a trade-off table by using larger voxel size and true performance (not just in the perception part but also other aspect).
> >
> > 2. license issue
> > I still do not feel safe about waymo license on second modification of its dataset. I check previously on the waymo license and it said the second modification of its dataset should be non-commercial-usage, inherits its original license (which means everyone use Occ3D-waymo should register on waymo website). Please take a care look into this.
> >
> > After all, the response solves most of my concern above and I may return to my original rating 5.

---

> > > ### Author Response · Authors · 2023-08-31
> > >
> > > Thank you for your continued engagement and for sharing your concerns. We would like to address the two issues you raised:
> > > 1. **nuScenes dataset:**
> > >     1.1. We have provided a visualization of the effects before and after employing ICP and ground fitting in the revised supplementary material, Figure 5. Figure 5.a shows the occupancy data generated directly, while Figure 5.b shows the generated data after applying the ICP algorithm and ground fitting techniques. This figure illustrates a scenario with a significant slope, resulting in the ground being divided into multiple layers. As can be seen from the figure, employing the ICP algorithm and ground fitting techniques successfully transforms the multi-layered ground into a single layer.
> > >
> > >     1.2. We are currently in the process of generating full comparison data of different voxel sizes. Unfortunately, this process requires more than a day to complete, and as a result, we will not be able to provide the table before the rebuttal deadline. However, we assure you that the trade-off table will be included in subsequent versions of the paper.
> > >
> > > 2. **License issue:**
> > >     Yes, we have mentioned this point in both the datasheet and the supplementary material. Users of our proposed Occ3D-Waymo must adhere to the Waymo dataset license, which prohibits commercial use.
> > >
> > > We hope this response addresses your remaining concerns, and we are grateful for your consideration to return to your original rating of 5.

---

### Official Review · Reviewer_PEmo · 2023-07-20
**Occ3D: Promising Dataset and Benchmark**

**Rating:** 9
**Confidence:** 4

**Strengths:**

1. Novel Dataset: The authors have provided a comprehensive and well-structured description of their dataset construction pipeline, offering valuable insights into the generation steps, and addressing the challenges encountered during the process. The level of detail provided aids in the understanding of their dataset’s structure and fosters reproducibility. This author believes the dataset is a valuable contribution to the research community.
2. Novel Network Architecture: The experimental results presented in the paper are well presented and show excellent performance. The proposed CT F-Occ model consistently outperforms most other algorithms on both datasets created by the authors. The impressive performance of the CTF-Occ model demonstrates its effectiveness and superiority in 3D occupancy prediction, reaffirming the significance of the proposed approach in advancing the state-of-the-art in this area.

**Additional Feedback:**

This paper exhibits a commendable level of readability and clarity, rendering it accessible to a broad audience. The datasets created by the authors showcase a high standard of quality, bolstering the reliability and relevance of their research. The proposed model's exceptional performance underscores its significance and potential contribution to the field of 3D occupancy prediction. By presenting both well-structured datasets and a compelling model, the authors have made a notable advancement in the domain, fostering valuable insights and advancements in this area of study. I recommend an accept.

**Clarity:**

The paper is very well written. Figures are illustrative and very helpful in understanding the label generation process and network architecture.

**Correctness:**

I believe all claims are correct and the evaluation methods and experiment design appropriate and performed correctly.

**Documentation:**

I believe that the documentation is sufficient and that there are no concerns about availability and maintenance of the dataset.

**Ethics:**

I do not suspect there are any ethical concerns with the submission.

**Limitations:**

The authors discuss limitations of their label generation pipeline. I would be interested to see a short discussion of limitations with regards to their occupancy prediction network.

**Opportunities For Improvement:**

1. Broader Impacts: It would have been nice to see an explicit discussion of broader impacts of the work. The proposed dataset and occupancy network will likely have the most impact for development of autonomous driving algorithms, the authors could include a discussion of how their dataset could be used for positive or negative societal impact in this area.
2. Comparison to Existing Datasets: To provide some additional comparison to other approaches in occupancy prediction, the authors could adapt their model to perform scene completion using an established scene dataset with occupancy labels, e.g. ScanNet. Demonstrating scene completion results on an established dataset would help contextualize the performance of their approach on more well-studied occupancy prediction problems (i.e. shape/scene completion).

**Relation To Prior Work:**

I believe the related work is sufficient.

**Summary And Contributions:**

The authors have made significant contributions to the field of 3D occupancy prediction by introducing and publicly sharing a two-part dataset: Occ3D-nuScenes and Occ3D-Waymo. These datasets are valuable resources for researchers, with Occ3D-nuScenes comprising 40,000 frames and covering 17 different object classes and Occ3D-Waymo comprising of 200,000 frames encompassing 15 object classes. The authors have also developed a novel transformer-based model called the Coarse- to-Fine Occupancy (CTF-Occ) network. The CTF-Occ network showcases remarkable performance in 3D occupancy prediction, elevating the capabilities of existing methods in this domain.

---

> ### Author Response · Authors · 2023-08-27
> **Response to Reviewer PEmo**
>
> We appreciate the reviewer's comment. We are encouraged by the high rating score for our proposed dataset and the presented study.
>
> We have addressed each of the concerns separately below.
>
> **1. Discussion of societal impact**
> *Original comment:
> The authors could include a discussion of how their dataset could be used for positive or negative societal impact in this area.*
>
> ***Response:***
> The proposed Occ3D serves as a large-scale, high-quality benchmark for 3D occupancy prediction and has been employed in recent research on occupancy prediction task. Furthermore, we hosted a challenge at CVPR 2023 to enhance the popularity and utilization of this dataset within the community.
>
> **2. Limitations of network**
> *Original comment:
> The authors discuss limitations of their label generation pipeline. I would be interested to see a short discussion of limitations with regards to their occupancy prediction network.*
>
> ***Response:***
> We identify several limitations of the proposed model, which also suggest directions for future research:
> * The current model recognizes only limited categories. Enhancing its ability to work with an open-vocabulary system would substantially improve its functionality.
> * At present, the model utilizes only three frames of images. Exploiting longer temporary information could lead to further performance enhancements.
> * Optimizing the network to reduce both memory consumption and computational cost would make the model more efficient and potentially more applicable.

---

### Official Review · Reviewer_EF4V · 2023-07-21
**Clear motivation, supporting evidence for their claims**

**Rating:** 8
**Confidence:** 3
**Correctness:** No problems at all.
**Clarity:** Well written. Straightforward approac…

**Strengths:**

1. Comprehensive Benchmark: The proposed benchmark provides a comprehensive evaluation framework for 3D occupancy prediction. By incorporating three distinct stages (voxel densification, occlusion reasoning, and image-guided voxel refinement), it covers various aspects of the prediction process, allowing for a more thorough analysis of different models.
2. Diverse and Real-world Datasets: The introduction of two new datasets, Occ3D-Waymo and Occ3D-nuScenes, derived from Waymo Open Dataset and nuScenes Dataset, respectively, enhances the diversity and real-world relevance of the evaluation. These datasets enable the evaluation of 3D occupancy prediction models on complex and dynamic scenes, closer to real-world driving scenarios.
3. Clear Explanations: The authors' writing style is clear and concise, effectively conveying the ideas and concepts involved in 3D occupancy prediction. The detailed explanations of each step in the generation pipeline make it easier for readers to understand the methodology and reproduce the results.
4. Straightforward Implementation: The ideas presented in the paper are straightforward and easy to follow, making it more accessible for researchers and practitioners to adopt and build upon the proposed methods. The simplicity of the approach might lead to faster adoption and integration in related research areas.
5. State-of-the-art Performance: The proposed CFT-Occ network outperforms previous 3D occupancy prediction models. By incorporating cross-attention and employing an efficient coarse-to-fine strategy to aggregate 2D image features into 3D space, it achieves state-of-the-art results, highlighting its effectiveness and potential for practical applications.
6. They provide a methodology to verify their datasets' quality. Moreover, their proposed method's superior performance indicates their proposed dataset has a great direction for future research.

**Additional Feedback:**

In reference, there are inconsistency for the conference name.
ex)
[4] Holger Caesar, Varun Bankiti, Alex H Lang, Sourabh Vora, Venice Erin Liong, Qiang Xu, Anush Krishnan,
Yu Pan, Giancarlo Baldan, and Oscar Beijbom. nuscenes: A multimodal dataset for autonomous driving. In
Proceedings ofthe IEEE/CVF conference on computer vision and pattern recognition, pages 11621–11631,
2020.

[17] Alex H Lang, Sourabh Vora, Holger Caesar, Lubing Zhou, Jiong Yang, and Oscar Beijbom. PointPillars:
Fast Encoders for Object Detection from Point Clouds. In CVPR, pages 12697–12705, 2019.
322

If the authors make them consistent, the paper's quailty would be improved.

**Documentation:**

Clear documentation.

**Ethics:**

Well elaborated. But, I'm slightly worried about that some of the data might include unwanted captures. This is not really a main consideration of this work but is for the autonomous driving dataset.

**Limitations:**

See the weakness part. Although the proposed dataset generation steps rely on heuristic algorithms, future work might address this issue.

**Opportunities For Improvement:**

The proposed method for 3D occupancy prediction prominently relies on the utilization of heuristic algorithms, which, while effective to a certain extent, necessitates further exploration in future research to identify and devise more sophisticated and improved ways to accurately label occlusion, thus enhancing the overall predictive performance. Furthermore, the authors are encouraged to include a detailed analysis of failure cases encountered by the proposed algorithm in order to present a comprehensive understanding of its limitations and potential shortcomings, which could serve as invaluable insights for refining and advancing future iterations of the approach. Additionally, in Figure 3, it is advisable for the authors to consider employing distinct and discernible patterns in each color used, especially taking into consideration the possibility that some readers might print the paper in monochromatic white-black format, ensuring optimal visual comprehension and accessibility of the essential information conveyed in the figures.

**Relation To Prior Work:**

Clearly discussed.

**Summary And Contributions:**

Robotic perception necessitates comprehending the 3D geometry and semantics of the surroundings. Currently, prevalent methods primarily focus on approximating 3D bounding boxes, disregarding finer geometric details and facing challenges with unfamiliar objects. In response to these limitations, a novel task called 3D occupancy prediction has emerged, with the goal of precisely estimating occupancy states and semantics in a scene. To support this task, the authors of the research have devised a label generation pipeline that generates dense, visibility-aware labels for any given scene. This pipeline comprises three stages: voxel densification, occlusion reasoning, and image-guided voxel refinement. To assess the effectiveness of their approach, the researchers established two benchmarks, namely Occ3D-Waymo and Occ3D-nuScenes, utilizing data from the Waymo Open Dataset and the nuScenes Dataset, respectively. They conducted an extensive analysis using various baseline models on the proposed dataset. Additionally, they introduced a new model called the Coarse-to-Fine Occupancy (CTF-Occ) network, which exhibited superior performance on the Occ3D benchmarks.

---

> ### Author Response · Authors · 2023-08-27
> **Response to Reviewer EF4V**
>
> We appreciate the reviewer's comment. We are encouraged by the high rating score for our proposed dataset and the presented study.
>
> We have addressed each of the concerns separately below.
>
> **1. Detailed analysis of failure cases**
> *Original comment:
> Furthermore, the authors are encouraged to include a detailed analysis of failure cases encountered by the proposed algorithm in order to present a comprehensive understanding of its limitations and potential shortcomings, which could serve as invaluable insights for refining and advancing future iterations of the approach.*
>
> ***Response:***
> Thanks for the suggestion. Please see the analysis of failure cases in the comment "Author Response to Common Issues". We will update this part  in the revised version of our paper.
>
> **2. Employing distinct and discernible patterns**
> *Original comment:
>  Additionally, in Figure 3, it is advisable for the authors to consider employing distinct and discernible patterns in each color used, especially taking into consideration the possibility that some readers might print the paper in monochromatic white-black format, ensuring optimal visual comprehension and accessibility of the essential information conveyed in the figures.*
>
> ***Response:***
> Thanks for the suggestion. We will update this part in the revised version of our paper.
>
>
> **3. Inconsistency for the conference name**
> *Original comment:
> In reference, there are inconsistency for the conference name.*
>
> ***Response:***
> Thanks for pointing this. We have corrected this issue in the revised paper.

---

### Official Review · Reviewer_Dt1s · 2023-07-21
**Review of Occ3D**

**Rating:** 9
**Confidence:** 5
**Clarity:** The paper is well-written and easy to…

**Strengths:**

The paper has the following strength:

1. The new dataset is built on Waymo and NuScene raw sequences, which provide diverse and large amount of driving scenes.
2. The label generation process is carefully designed to reconstruct the actual geometry of the driving scenes while considering the occlusion reasoning.
3. The authors design matric to quantitatively ablate the necessity of each component in the labeling procedure.
4. The authors present their own state-of-the-art architecture tailored for their new benchmark.


**Additional Feedback:**

The feedback was provided in the above sections.

**Correctness:**

The authors provide a well-established dataset with adequate experiments on the new benchmark. They also include the quality check report for the new dataset.

**Documentation:**

The dataset is accessible to the public with documentation. The authors claim to release the development kit and related documentation soon.

**Limitations:**

Table 2 (row 1 to row2) shows that the proposed multi-frame aggregation process seems to be not effective for dynamic objects (vehicles, bicyclists and pedestrians).

**Opportunities For Improvement:**

The label-generation process considers consistency between the image domain and the LiDAR domain. The consistency heavily relies on sensor calibration. Have the authors considered camera-LiDAR self-calibration to adjust the sensor extrinsics during the label generation process besides image-guided voxel refinement?

**Relation To Prior Work:**

The authors discuss related literature in the paper and make a major contribution to addressing the issue of the previous 3D semantic scene completion task.

**Summary And Contributions:**

In autonomous driving, traditional approaches to 3D perception have primarily focused on 3D object detection and tracking, which often represent objects using 3D bounding boxes. However, such a representation restricts the capacity of neural networks to accurately perceive the actual geometry of driving scenes, potentially limiting their ability to find out-of-vocabulary and irregularly shaped objects. To address this limitation and provide a more comprehensive and semantically rich representation of 3D space, the authors introduce Occ3D—a novel benchmark for 3D occupancy prediction.

The authors precisely define the task of occupancy prediction as the ability to infer the occupied state of each voxel, along with its semantics, given the surrounding images.

The authors' contributions are as follows:

They present the first large-scale 3D occupancy prediction benchmark capable of occlusion reasoning for image-based perception.
They provide a detailed data generation process.
They conduct rigorous experiments to showcase the research potential their proposed benchmark can open up.

---

> ### Author Response · Authors · 2023-08-27
> **Response to Reviewer Dt1s**
>
> We appreciate the reviewer's comment. We are encouraged by the high rating score for our proposed dataset and the presented study.
>
> We have addressed each of the concerns separately below.
>
> **1. Consider camera-LiDAR self-calibration**
> *Original comment:
> Have the authors considered camera-LiDAR self-calibration to adjust the sensor extrinsics during the label generation process besides image-guided voxel refinement?*
>
> ***Response:***
> Existing datasets are meticulously calibrated between sensors, demonstrating that LiDAR points can be accurately mapped to image pixels through the provided projections, as referenced in Figure 5 in [A]. The topic of camera-LiDAR self-calibration is indeed intriguing and may offer significant benefits, especially for custom datasets.
> [A] Sun, Pei, et al. "Scalability in perception for autonomous driving: Waymo open dataset." Proceedings of the IEEE/CVF conference on computer vision and pattern recognition. 2020.
>
>
> **2. Multi-frame aggregation process**
> *Original comment:
> Table 2 (row 1 to row2) shows that the proposed multi-frame aggregation process seems to be not effective for dynamic objects (vehicles, bicyclists and pedestrians).*
>
> ***Response:***
> Thanks to the reviewer for highlighting this detail. In Table 2, the multi-frame aggregation (MFP) is adopted by default for dynamic objects in both row1 and row2. We further conducted an ablation study on the MFP by removing it, specifically for dynamic objects. The resulting indicators are as follows:
> | |  | MFP | | |  | SFP|  |
> | :----: | :----: | :----: | :----: |:----: | :----: | :----: | :----: |
> |  | IOU | Recall | Precision || IOU | Recall | Precision |
> | vehicle |  37.89 | 40.02  | 87.48  ||  5.87 | 8.53 | 95.38 |
> | bicyclist | 37.99  | 58.77  | 51.79  || 5.12  | 6.61  | 58.66 |
> | pedestrian | 28.25  | 37.21  | 53.98  || 3.65  | 4.81 | 60.13 |
> | sign | 12.57  | 14.80  | 45.45  || 3.47  | 3.85  |  61.44 |
>
> It can be observed that the MFP method has a substantial positive impact on the handling of dynamic objects.

---

> > ### Comment · Reviewer_Dt1s · 2023-08-30
> >
> > Thanks to the authors for their response and clarification. I will keep my rating.

---

### Official Review · Reviewer_dHUH · 2023-07-26
**Occ3D: A Large-Scale 3D Occupancy Prediction Benchmark for Autonomous Driving**

**Rating:** 6
**Confidence:** 4
**Clarity:** The paper is well-written and easy to…

**Strengths:**

The new Occ3D benchmark provides a way to evaluate occupancy prediction models more accurately. The supplementary material contains many details to implement the proposed pipeline that builds dense 3D labels from LiDAR. The authors successfully hosted a challenge in conjunction with CVPR 2023, which shows a positive impact of the proposed benchmarks on the community.

**Additional Feedback:**

Even though the technical novelty is somewhat weak, the effort on building the new benchmark and the detailed information provided in the documentation must be very useful to many researchers in our community.

**Correctness:**

The proposed pipeline based on voxel densification by accumulating LiDAR frames and careful refinement technically sounds. Although the paper does not contain the evaluation protocol and metrics, but the documentation of the challenge has enough information and the experiments performed correctly.

**Documentation:**

Sufficient detail on the proposed benchmark is provided. Since the authors hosted a challenge with this benchmark, there is already a good documentation on how to use this benchmark.

**Limitations:**

The limitations are mentioned in the paper. However, since the proposed benchmark is built by an algorithm, there might be some errors. If so, discussing about the errors will be very helpful,

**Opportunities For Improvement:**

- Missing details for Table 2. The evaluation setting is not given. If the ground truth used in Table 2 is from the proposed pipeline, showing the best performance on the same setting the model is trained on cannot justify the effectiveness/usefulness of the design choices.

- Several important pieces of information are not provided in the main paper, such as the evaluation protocol. Although the missing information is provided in the supplementary/documentation, mentioning them in the main paper would be great.


**Relation To Prior Work:**

In related work section, relevant tasks are introduced such as 3D detection, occupancy grid mapping, and semantic scene completion. However, as there were several methods to build dense label from LiDAR data before [a,b,c], it would be better to discuss the missing part in the previous methods to reveal its novelty of the proposed pipeline.

[a] Instant Domain Augmentation for LiDAR Semantic Segmentation, CVPR 2023
[b] Complete & Label: A Domain Adaptation Approach to Semantic Segmentation of LiDAR Point Clouds, CVPR 2021
[c] Domain Transfer for Semantic Segmentation of LiDAR Data using Deep Neural Networks, IROS 2020

**Summary And Contributions:**

This work presents a dense label-generation pipeline consisting of voxel densification, occlusion reasoning, and image-guided voxel refinement. Using the pipeline, two benchmarks for 3D occupancy prediction task are also introduced; Occ3D-Waymo and Occ3D-nuScenes. Also, a new occupancy prediction model, named Coarse-to-Fine Occupancy (CTF-Occ) network, is proposed, which outperforms recent occupancy prediction models on Occ3D benchmarks.

---

> ### Author Response · Authors · 2023-08-23
> **Response to Reviewer dHUH**
>
> Thank you for your careful analysis of our work. We hope the following response addresses your concerns.
>
>
> **1.Setting and effectiveness of Table2**
> *Original comment:
> Missing details for Table 2. The evaluation setting is not given. If the ground truth used in Table 2 is from the proposed pipeline, showing the best performance on the same setting the model is trained on cannot justify the effectiveness/usefulness of the design choices.*
>
> Response:
> **Setting:**
> In Section 4.1, we describe the evaluation setting in detail. Specifically, we further define the 2D ROI to filter the pixel region and propose the 3D label query method to determine corresponding 3D voxel labels. Regarding the semantic label of the image annotation as ground truth, and the semantic label of the projected voxel as prediction, we adopt the standard Precision, Recall, IoU and mIoU metric.
>
> **Effectiveness:**
> The performance mentioned in Table 2 does not represent the model's performance but rather the 3D-2D consistency. Table 2 is utilized to evaluate the quality of the dataset and to ensure the effectiveness of design choice in our dataset generation pipeline.
> In Section 4.2, we justify the effectiveness of our design choices, including multi-frame aggregation, mesh reconstruction, and image-guided voxel refinement. By adopting these design choices, we significantly enhance the consistency metric.
> If we do not accurately respond to your comment, please revise it and provide additional description.
>
>
> **2.Recommendation for information inclusion in the main paper**
> *Original comment:
> Several important pieces of information are not provided in the main paper, such as the evaluation protocol. Although the missing information is provided in the supplementary/documentation, mentioning them in the main paper would be great.*
>
> Response:
> Thank for the feedback. We acknowledge the importance of the mentioned details, especially the evaluation protocol, being present in the main paper for clarity and ease of reference. The primary reason some of these details were placed in the supplementary material was due to the page limitation of the main paper. In the revised version, we will incorporate the essential information into the main paper to ensure its comprehensiveness. We appreciate your valuable suggestion.
>
> **3.Discussion  about the errors**
> *Original comment:
>  However, since the proposed benchmark is built by an algorithm, there might be some errors. If so, discussing about the errors will be very helpful.*
>
> Response:
> We discuss the errors and future approach to reduce these errors in the comment **"Author Response to Common Issues"**.

---

### Author Response · Authors · 2023-08-23
**Author Response to Common Issues.**

We greatly appreciate the time and effort of all reviewers, as well as their constructive feedback and valuable suggestions.

Before we introduce our specific response to the review comments, let us list the two common issues in the review comments:

**(Issue #1) Analysis of errors and future itearations (by 2 reviewers: dHUH and EF4V).**
Response:

We have carefully designed the dataset generation pipeline and significantly enhanced the dataset quality, as shown in Table 2. The quality can be further improved in several ways:
1. Sensor Calibration Error: Since we use LiDAR scans to construct high-quality occupancy labels for camera perception, the calibration between LiDAR and cameras becomes critical. Conducting multi-frame aggregation also relies on precise sensor calibration.
2. Dynamic and Deformable Objects: For dynamic objects, we extract the points located within the box and aggregate them. However, some dynamic objects may not have box annotations, such as running animals, and some objects may not satisfy the rigid body assumption, like a person swinging their arms. There will be motion blur problems in these cases.
3. General Objects: Both the nuScenes and Waymo datasets only annotate limited categories. Out-of-vocabulary objects such as trash cans and traffic cones are all regarded as general objects. Further human annotation to provide fine-grained details will help in reproducing an intelligence with unbounded understanding and benefit auto-driving research.


**(Issue #2) Related work update (by 3 reviewers: dHUH,EF4V and S7Q7).**
Response:

We are grateful to the reviewers for their valuable comments and insights. We will incorporate their suggestions by updating the related work and correcting the typos identified.

---

### Decision · Program_Chairs · 2023-09-22

**Decision:**

Accept (Poster)

**Comment:**

This paper contributes a pipeline for data generation of 3D occupancies and a resulting benchmark dataset in the context of autonomous driving (Occ3D).  The reviewer responses were mostly positive, recognizing the value of the pipeline and dataset, and the thoroughness of the experimental results and the paper writeup.  One negative reviewer expressed concerns, primarily regarding some gaps in the discussion of prior work and technical details of how the contributed dataset builds upon prior work.  These and other identified weaknesses were addressed by the authors in the rebuttal and discussion period.  Overall, the positive reviewers kept their ratings and supported acceptance of the paper.  The initially negative reviewer engaged with the authors in discussion but did not revise their score or express an opinion against the paper being accepted.  The AC thus finds no basis to overrule the majority reviewer opinion and recommends that the paper be accepted.